

# Detectability of seismic waves from the submarine landslide that caused the 1998 Papua New Guinea tsunami

Akio Katsumata[1], Yasuhiro Yoshida[2], Kenji Nakata[1], Kenichi Fujita[1], Masayuki Tanaka[1],
Koji Tamaribuchi[1], Takahito Nishimiya[1], and Akio Kobayashi[1]

[1]Meteorological Research Institute, Japan Meteorological Agency, 1-1 Nagamine, Tsukuba, Ibaraki Prefecture, Japan
[2]Meteorological College, Japan Meteorological Agency, 7-4-81 Asahi-cho, Kashiwa, Chiba Prefecture, Japan

**Correspondence:** Akio Katsumata (akatsuma@mri-jma.go.jp)

**Abstract.** On 17 July 1998, a tsunami caused serious damage on the northern coast of Papua New Guinea about 20 min after
the mainshock of an $M_w$ 7.0 earthquake. The tsunami has been attributed to a submarine landslide that occurred about 13 min
after the mainshock because its arrival at the coast was too late and its height too great to be the direct result of the fault slip of
the earthquake. Bathymetric data recorded after the tsunami revealed an amphitheater-like structure that was consistent with a

recent submarine landslide. Most current tsunami warning systems are based on analysis of the early arrivals of seismic waves
generated by an earthquake. In this study we investigated whether evidence of the landslide could be identified in the coda
waves recorded after the mainshock. Based on previous studies of the tsunami source, we constructed synthetic seismograms to
represent the submarine landslide and compared them to the observed coda waves of the preceding earthquake, with particular
attention to the period around 13 min after the mainshock. We found phases possibly corresponding to the landslide event.

However, they were easily covered with coda waves from the mainshock. We concluded that the 1998 landslide was too small
to be evident in the coda waves following the magnitude 7 earthquake.

*Copyright statement.* TEXT

## 1   Introduction

A tsunami struck the north coast of Papua New Guinea (PNG) on 17 July 1998. The height of the tsunami was more than

10 m and its passage resulted in the deaths of more than two thousand people (e.g., Kawata et al. , 1999; Tappin et al. ,
1999; Synolakis et al. , 2002). The tsunami struck the coast about 20 min after an earthquake (Kawata et al. , 1999) of $M_w$
7.0 (GCMT, Dziewonski et al. , 1981; Ekström et al. , 2012). The arrival time of the tsunami was too late and its height too
great for it to have been a direct result of the fault slip of the earthquake. Moreover, bathymetric data recorded after the
earthquake revealed an amphitheater-like structure consistent with recent submarine slumping about 20 km off the coast

(Tappin et al. , 1999). It was therefore presumed that the tsunami was caused by a submarine landslide (e.g., Tappin et al. ,
1999; Heinrich et al. , 2000; Synolakis et al. , 2002). Numerical simulations of the landslide (e.g., Tappin et al. , 1999; Watts et al. ,



2003; Lynett et al. , 2003) provided landslide masses of 6.4 km$^3$ (Tappin et al. , 1999), 4 km$^3$ (Heinrich et al. , 2000), and 9 km$^3$ (Watts et al. , 2003).

Early warnings of impending tsunamis are usually issued based on analyses of the early arrivals of seismic waves generated by earthquakes. If the submarine landslide after the 1998 PNG earthquake had been detected in the seismic data recorded immediately after the earthquake, it might have been possible to provide advanced warnings to PNG residents of the imminent tsunami. In this study, we investigated whether the seismic signature of the submarine landslide was present in the coda waves of the seismic data recorded after the mainshock of the 1998 earthquake.

## 2 Observed seismic records

The broadband seismic data recorded during the $M_w$ 7.0 earthquake that preceded the 1998 tsunami has been archived at both the Earthquake Research Institute (ERI; The University of Tokyo) and the Incorporated Research Institutions for Seismology (IRIS; headquarters in Washington DC). The two seismic stations from which data were mainly used in this study were 146 km (Jayapura station, JAY) and 918 km (Port Moresby station, PMG) from the epicenter (Fig. 1).

Synolakis et al. (2002) calculated that the submarine landslide occurred 13 min after the mainshock on the basis of hydrophone data recorded at Wake Island. Heidarzadeh and Stake (2015) used tide gauge data to estimate that the landslide occurred 12–17 min after the mainshock. Synolakis et al. (2002) pointed out that seismic signal corresponding to the landslide could not be recognized at PMG. Records at JAY show no significant events at around 13 min after the mainshock in the four frequency bands considered (Fig. 2). Whereas any significant events are not seen in the records at six stations around source area (Fig. 3), small phases are recognized about 13 min after the S-arrival in the 50–100 s band of horizontal components at JAY and PMG (bold parts of seismic waves in Fig. 3). The particle motions of the bold parts were oriented to north-south at JAY and northeast-southwest at PMG, which were roughly consistent with those of the Love waves from the source area. However, there were not such events in the records at the other stations (Fig. 3).

## 3 Comparison of synthetic seismograms representing the 1998 submarine landslide with coda waves from the 1998 earthquake

We estimated the amplitudes of the seismic waves generated by the 1998 landslide on the basis of the work of Watts et al. (2003), as described below. Relatively large amplitudes were expected based on the model by Watts et al. (2003) because of its relatively large assumed mass.

When a landslide starts, the change in the force that has supported the landslide mass generates seismic waves. The force ($F$) generated by a landslide is defined by $F = ma$, where $m$ is the mass of the landslide and $a$ is its acceleration. Watts et al. (2003) assumed the landslide mass to be a half ellipsoid and, on the basis of the parameters given in Table 1, estimated its volume to be 9 km$^3$ ($\frac{\pi wbT}{6}$) and its mass to be $19 \times 10^{12}$ kg. Using this mass and the initial acceleration $a_0$ (0.36 m s$^{-2}$; Table 1) of Watts et al. (2003) gives an estimated force of $7 \times 10^{12}$ N. The volume determined by Watts et al. (2003) (9 km$^3$) is





larger than those of Heinrich et al. (2000) (4 km$^3$), Synolakis et al. (2002) (4 km$^3$), and Tappin et al. (2008) (6.4 km$^3$), and the acceleration of 0.36 m s$^{-2}$ is smaller than that used by Tappin et al. (2008) (0.47 m s$^{-2}$). The force and volume estimated here for the 1998 PNG landslide are comparable to those estimated by Kanamori and Given (1982) for the landslide associated with the 1980 Mount St. Helens eruption ($10^{13}$ N and 2.5 km$^3$, respectively).

The time history of a landslide must be assumed in order to estimate the amplitude of the resultant seismic wave. Watts et al. (2003) assumed a characteristic time of 32 s. Tappin et al. (2008) estimated that the sliding process lasted about 100 s. We assumed that the sliding process was completed within several tens of seconds with an initial acceleration stage followed by a deceleration stage (Fig. 4), and that the total impulse would not have been balanced because part of the deceleration stage would be affected by interaction of the sliding mass with sea water. The curves for each of the four intervals defined on Fig.

4 were expressed as trigonometric functions (Table 2). Five different time histories were considered (Table 3). The peaks of $\int F\mathrm{d}t/m$ range from 3 to 11 m s$^{-1}$. The value 11 m s$^{-1}$ of case (c) (Table 3) is close to $u_{max}$ of Table 1. We constructed synthetic seismic records (Figs. 5 and 6) by applying the method of Takeo (1985) and using the seismic velocity model given in Table 4.

The amplitudes of of our synthetic records are comparable to or larger than those of the seismic waves recorded after the

mainshock of the $M_w$ 7.0 earthquake in the 50–100 s passband (Figs. 5 and 6). The phases indicated with bold lines could be the seismic waves from the landslide which caused the disastrous tsunamis. Whereas the ratios of the observed amplitudes to the synthetics at JAY were generally small, those at PMG were not so small. It is difficult to retrieve a consistent result about the source parameters from this comparison. The amplitudes of the phase was smaller than the successive seismic wave at JAY. It is unlikely to recognize the occurrence of the landslide based on the seismic records after the large earthquake.

## 4   Discussion

Several studies have shown that seismic waves generated by landslides can be detected. The landslide associated with the 1980 eruption of Mount St. Helens (Kanamori and Given , 1982; vol. 2.5 km$^3$) is the largest of these. Yamada et al. (2012) analyzed seismic waves generated by landslides (vols. of up to $13.6 \times 10^6$ m$^3$) caused by heavy rain in Japan. Li et al. (2017) used seismic data to analyze the dynamic process of a landslide (vol. $5 \times 10^6$ m$^3$) in southwest China. Landslides in Greenland

that caused tsunamis in 2000 (Dahl-Jensen et al. , 2004; underwater vol. $30 \times 10^6$ m$^3$) and 2017 (Chao et al. , 2018; vol. 35–51 $\times 10^6$ m$^3$) were recorded by seismometers. For the 2000 Greenland event, Dahl-Jensen et al. (2004) estimated the surface-wave magnitude of the event to be 2.3.

In cases such as the 1998 PNG tsunami, where the submarine landslide that caused it occurred some minutes after an earthquake, the seismic signature of the landslide can be masked by the seismic waves generated by the earthquake. In 1908,

after a magnitude 7.1 earthquake, the southern Italian coast was struck by a 5–10 m tsunami (Salamon et al. , 2011) that Billi et al. (2008) attributed to a submarine landslide. The event in Italy in 1908 appears to be similar to the 1998 PNG earthquake and tsunami.



Because some onshore earthquakes of magnitude about 7 or greater are known to have caused landslides, it is likely that offshore earthquakes of similar magnitude cause submarine landslides. Kodaira et al. (2012) suggested that a submarine landslide might have been caused by the 2011 Tohoku-oki earthquake ($M_w$ 9.0). However, the seismic and tsunami signatures of such landslides are likely masked by the responses to the fault motions of the earthquakes. An extraordinarily large landslide

mass would be required to prevent its signatures from being overwhelmed by the responses of the fault motion of a magnitude 7 or greater earthquake.

It is considered that long periods of seismic inactivity or quick sedimentation (Sawyer et al. , 2017) is required for the accumulation on submarine slopes of sufficient sediment to generate a landslide large enough to cause a tsunami of comparable size to the 1998 PNG tsunami. The Mediterranean is a seismically inactive region where, according to Salamon et al. (2011),

most of the known tsunamis have been caused by submarine landslides. Because southeast Asia is a seismically active area, such an inactive condition unlikely realizes. It is plausible that the heavy rainfall characteristic of the equatorial belt promotes rapid accumulation of large volumes of marine sediment that might contribute to the occurrence of submarine landslides, including the 1998 PNG landslide, despite their location in a seismically active region. A few tsunamis in Japan, where tectonic activity is very high, have also been attributed to submarine landslides (e.g., Baba et al. , 2012; Baba et al. , 2017).

Chao et al. (2018) considered the use of seismic records to detect landslides in order to provide early warnings of impending tsunamis. However, as we have demonstrated here, the seismic signature of large submarine landslides can be overwhelmed by the seismic coda waves generated by earthquakes, so it can be difficult to detect submarine landslides soon after large earthquakes by this method. An alternative method of providing early warnings for tsunamis generated by submarine landslides is needed. Monitoring by networks of ocean-bottom tsunami gauges such as those of DONET (Kawaguchi et al. , 2015) or S-

net (Uehira et al. , 2016) might be a useful approach. For the 1998 PNG event, the tsunami source was about 20 km from the shoreline. This distance is shorter than the 30 km spacing between S-net sensors, so an array of similar dimensions to the S-net system would be too sparse for direct detection of tsunami waves such as those of the 1998 PNG tsunami with plural sensors. Other technologies with potential for direct detection of tsunami waves are tsunami radar (Barrick , 1979) and a fine barometer (Arai et al. , 2011), with which a tsunami is detected by reflection of electromagnetic waves or atmospheric pressure changes.

## 25  5  Conclusions

We investigated whether the tsunami-generating submarine landslide that occurred about 13 min after the 1998 PNG earthquake could be identified in the coda waves of the seismic data. We constructed synthetic seismograms to represent the seismic signature of the landslide and compared them to the seismic data recorded after the earthquake, with particular attention to the period around 13 min after the earthquake. We found small seismic phases possibly from the landslide. However, those phases

were difficult to be recognized as an indication of a disastrous tsunami. Other methods are needed to provide data for early warnings of tsunamis generated by submarine landslides of similar size (or smaller) to the one that generated the 1998 PNG tsunami. Networks of ocean-bottom tsunami gauges, similar to those provided by the DONET and S-net arrays in Japan, are among the likely candidates for this approach.





*Data availability.* Seismic data from the Jayapura seismic station are available at http://ohpdmc.eri.u-tokyo.ac.jp/. Seismic data from other seismic stations shown in this study are available at https://www.iris.edu/hq/.

*Author contributions.* AKa analyzed the observed and synthetic seismic records and compiled most of the paper. YY suggested input to the methodology for construction of the synthetic records used in this study. KN researched previous studies of the 1998 PNG earthquake

5 and tsunami. KF undertook preliminary research on the observed seismic records. MT, KT, TN, and AKo contributed to the research into previous studies of tsunamis caused by landslides and participated in related discussions.

*Competing interests.* The authors declare that they have no conflicts of interest directly relevant to the content of this article.

*Acknowledgements.* We used seismic data archived at the Earthquake Research Institute (The University of Tokyo) and at the Incorporated Research Institutions for Seismology. We used a computer program developed by Prof. Minoru Takeo to calculate synthetic records.



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





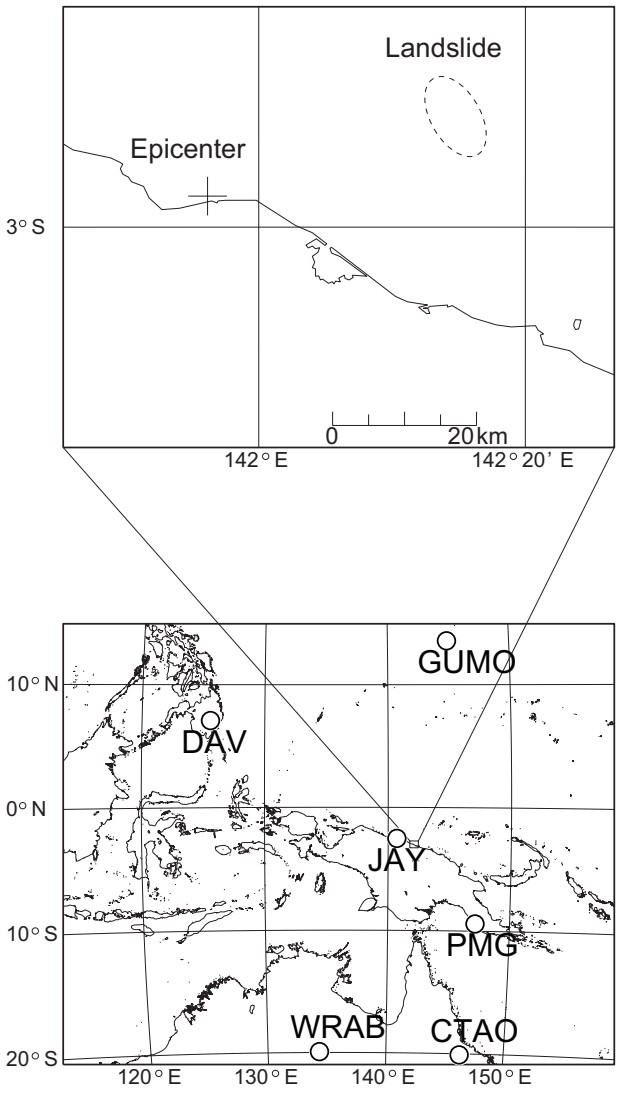

**Figure 1.** Maps of Papua New Guinea showing the location of the source area of the 1998 tsunami. The cross in the upper map indicates the epicenter of the $M_w$ 7.0 earthquake that occurred at 8:49 on 17 July 1998 (UTC) and the ellipse (broken line) marks the source location of the tsunami (as estimated by Tappin et al. , 1999) that struck the coast about 20 min after the mainshock. Circles on the lower map indicate the locations of seismic stations at Guam (GUMO), Davao (DAV), Jayapura (JAY), Port Moresby (PMG), Tennant Creek (WRAB), and Charters Towers (CTAO).



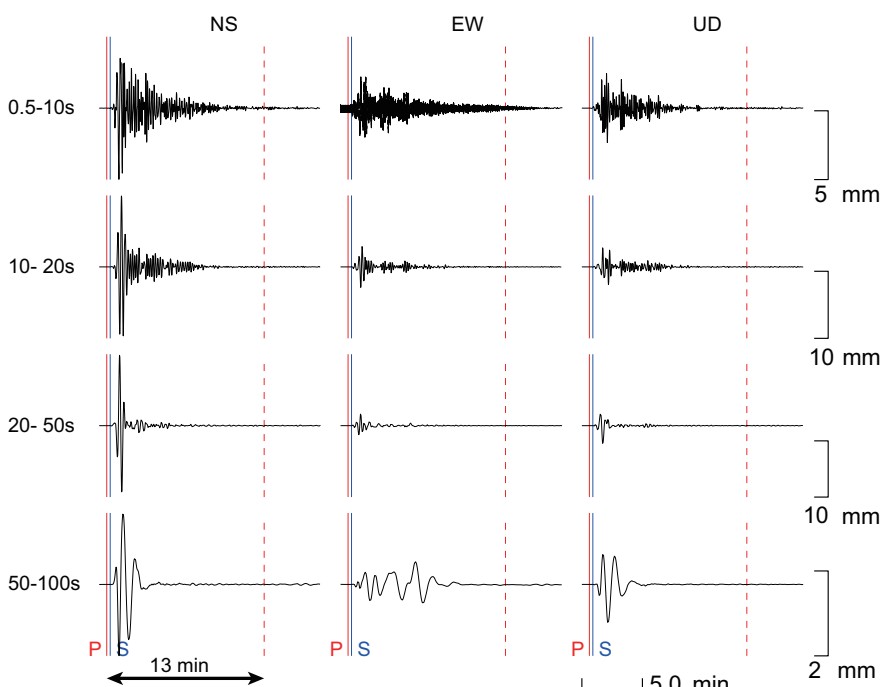

**Figure 2.** Seismic records obtained at station JAY (epicentral distance 146 km; location shown in Fig. 1) with four different passband filters applied after instrument response correction. Calculated P-wave (red lines) and S-wave (blue lines) arrivals for the 1998 earthquake are shown. The red broken line marks 13 min after the P-wave arrival.


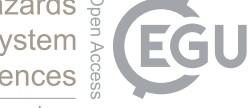

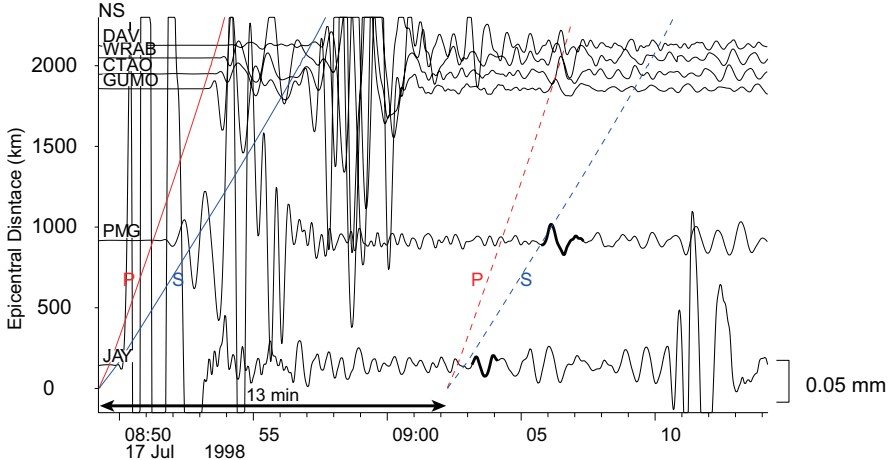

**Figure 3.** Seismic records of NS-component at six stations (Fig. 1) aligned according to the epicentral distance. A 50–100 s bandpass filter was applied to all records shown. Calculated P-wave (red line) and S-wave (blue line) arrivals for the mainshock are shown. The red and blue broken lines denote 13 min after the calculated arrivals which should indicate arrivals of seismic waves from the submarine landslide. The bold parts of seismic waves at JAY and PMG correspond to those in Figs. 5 and 6 and

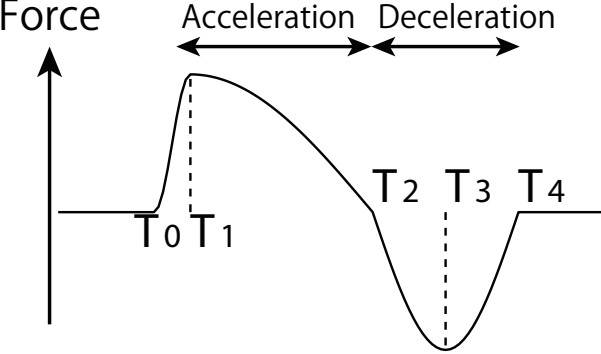

**Figure 4.** Schematic diagram showing the assumed source-time function for the force of the submarine landslide. Five different sets of intervals $T_1 - T_0$, $T_2 - T_1$, $T_3 - T_2$, and $T_4 - T_3$ were considered (Table 3).





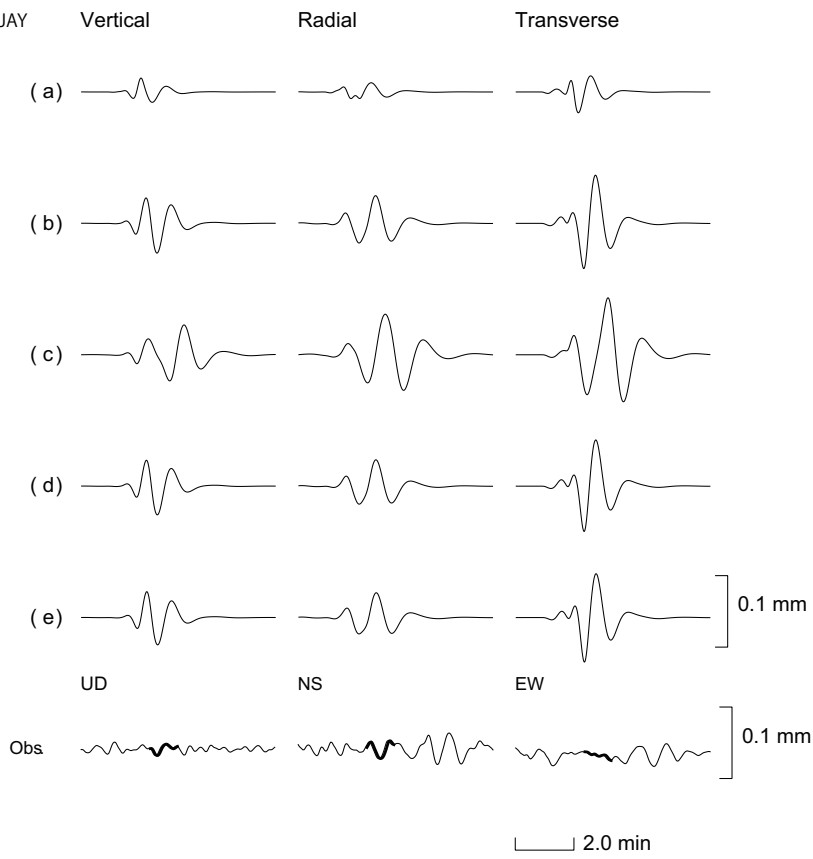

**Figure 5.** Synthetic seismic records constructed to represent the signature of the submarine landslide as it would have been recorded at station JAY, 176 km from the landslide source area (Fig. 1). The vertical and radial components are as viewed at an observation point along the assumed force direction. The transverse component is as viewed at an observation point on a line perpendicular to the direction of the source force. Different source-time functions were assumed for (a) to (e), as shown in Table 3. The seismic data recorded at station JAY 12.5 min after the mainshock are also shown (bottom panel). The bold parts of the observed records correspond to those in Figs. 3. A 50–100 s bandpass filter was applied to all records shown.



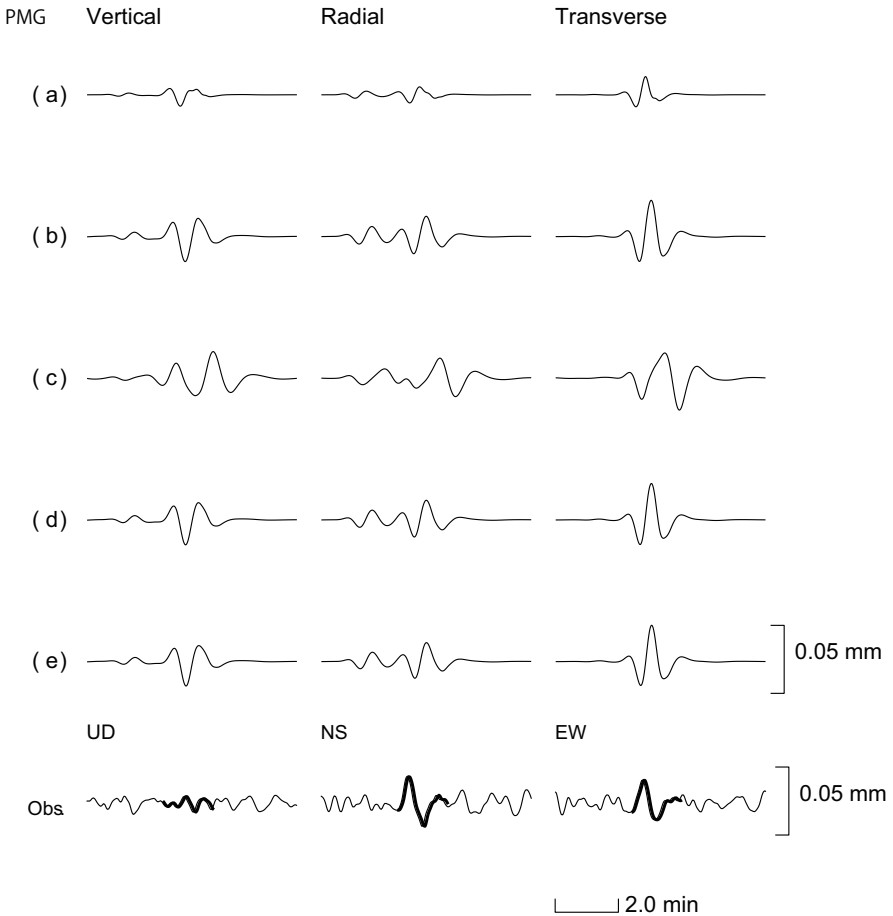

**Figure 6.** Synthetic seismic records as in Fig. 5, but for station PMG, 905 km from the landslide source area (Fig. 1).



**Table 1.** Parameters used by Watts et al. (2003) to simulate the 1998 PNG tsunami.

| Quantities | Values |
| :---: | :---: |
| $\gamma$ | 2.15 |
| $b$ (km) | 4.5 |
| $T$ (m) | 760 |
| $w$ (km) | 5 |
| $d$ (m) | 1500 |
| $\theta$ (degrees) | 12 |
| $a_o$ (m s$^{-2}$) | 0.36 |
| $u_{max}$ (m s$^{-1}$) | 11.6 |
| $s_o$ (km) | 375 |
| $t_o$ (s) | 32 |
| $\lambda_o$ (km) | 7.8 |
| $\eta_o$ (m) | -25 |

$\gamma$, the specific density; $b$, the initial landslide length; $T$, the maximum initial landslide thickness; $w$, the maximum landslide width; $d$, the mean initial landslide depth; $\theta$, the mean initial incline angle; $a_o$, the slump initial acceleration; $u_{max}$, the maximum slump velocity; $s_o$, the characteristic distance of slump motion; $t_o$, the characteristic time of slump motion; $\lambda_o$, the characteristic wavelength; $\eta_o$, the characteristic tsunami amplitude.



**Table 2.** Trigonometric source-time functions for each of the four intervals defined on Fig. 4.

| | |
|---|---|
| $T_0 - T_1$ | $-A[\cos\{\pi(t - T_0)/(T_1 - T_0)\} - 1]/2$ |
| $T_1 - T_2$ | $A\cos\{\pi(t - T_1)/(T_2 - T_1)/2\}$ |
| $T_2 - T_3$ | $-A\sin\{\pi(t - T_2)/(T_3 - T_2)/2\}$ |
| $T_3 - T_4$ | $-A[\cos\{\pi(t - T_3)/(T_4 - T_3)\} + 1]/2$ |

$A$ denotes the peak of the force.



**Table 3.** Five sets of durations (s) used for the four intervals of the source-time function shown in Fig. 4.

|     | $T_1 - T_0$ | $T_2 - T_1$ | $T_3 - T_2$ | $T_4 - T_3$ |
| --- | --- | --- | --- | --- |
| (a) | 2  | 10 | 4  | 4  |
| (b) | 4  | 20 | 8  | 8  |
| (c) | 8  | 40 | 16 | 16 |
| (d) | 8  | 16 | 8  | 8  |
| (e) | 12 | 12 | 8  | 8  |





**Table 4.** Velocity structure model used to calculate synthetic records. $Q_P$ and $Q_S$ are P- and S-wave attenuation coefficients, respectively.

| Depth(Top) | Velocity (P) | Velocity(S) | Density | $Q_P$ | $Q_S$ |
| --- | --- | --- | --- | --- | --- |
| km | km s$^{-1}$ | km s$^{-1}$ | g cm$^{-3}$ | | |
| 0 | 5.3 | 3.06 | 2.6 | 300 | 150 |
| 4 | 6.1 | 3.52 | 2.7 | 300 | 150 |
| 14.6 | 6.7 | 3.87 | 3.0 | 500 | 250 |
| 31.5 | 8.0 | 4.62 | 3.2 | 600 | 300 |