# Peer review of "Detectability of seismic waves from the submarine landslide that caused the 1998 Papua New Guinea tsunami"

_Natural Hazards and Earth System Sciences, 2018_

## Referee Comment (RC4)

*Emile A. Okal* (I waive anonymmity)

I was very disappointed by this paper. It presents succinct research, ignores previous work which clearly contradicts the argument, makes statements which border on being outrageous, and proposes warning measures which are nothing short of naive. Finally, its style is poor and it was not even proofread.

The paper should be rejected.

- The main argument of the paper is that there is no detectable seismic signature to the landslide which generated the catastrophic PNG tsunami of 17 July 1998. This statement directly contradicts the work of *Okal* [2003], in which I presented (on Figure 3) and discussed in detail the record of the landslide at the same station JAY allegedly studied by the authors. It is clear that they used the wrong (very low-frequency) filters, and thus missed the signal. They do not justify working in such inadequate frequency bands, and completely ignore the detailed analysis of seismic and hydroacoustic phases which went into my 2003 paper.

- I note on Page 4, Line 9 the statement "*The Mediterranean is a seismically inactive region*"! This is completely false. The USGS catalog contains 1132 events with at least one magnitude reaching 5 or greater for the period 1963–2015, between latitudes 30 and 45°N, longitudes −5 and 35°E, and depths 0 and 100 km...

  This factually wrong scientific statement takes an insulting societal tone when confronted to the memory of the thousands of victims of earthquakes in the Mediterranean Basin, documented since historic times.

**Those two very serious shortcomings suffice to warrant rejection of the paper.**

- It is wrong to use the reference to *Tappin et al*. [2008] to suggest that the slide underwent a "deceleration stage affected by interaction of the sliding mass with sea water". All submarine slides will feature such interaction. What was unique in the PNG slide was that it was stopped abruptly when it abutted against the opposite wall of the amphitheater in which it took place. All of this was explained in detail by *Synolakis et al*. [2002] and *Okal* [2003]; as mentioned above, the authors seem to ignore the latter paper, as they ignore the fundamental paper by *Sweet and Silver* [2003], who conducted the *in situ* discovery and study of the slide.

- The dynamics of the underwater PNG landslide and of the Mt. St. Helens one are totally different, given that the latter was caused by an atmospheric explosion, and reached velocities of 70 m/s (as documented from films) which cannot be sustained by underwater landslides.

- The proposal to densely instrument the seafloor in order to detect and identify in real time a landslide and issue a warning is naive in the context of the PNG tsunami, given that the whole process would have to be realized in a few minutes. Most of the casualties at Sissano resulted from the lack of an escape route: the residents were trapped on a narrow spit of land between the Bismarck Sea and Sissano Lagoon. The only survivors had managed to climb the few trees which were not uprooted. As such distances, the only reliable means of tsunami mitigation is proper planning (the village should not have been built on the spit), and in real-time, self-evacuation.

**References**

Okal, E.A., *T* waves from the 1998 Papua New Guinea earthquake and its aftershocks: Timing the tsunamigenic slump, *Pure Appl. Geophys.,* **160,** 1843–1863, 2003.

Sweet, S., and E.A. Silver, Tectonics and slumping in the source region of the 1998 Papua New Guinea tsunami from seismic reflection images. *Pure Appl. Geophys.,* **160,** 1945–1968, 2003.

---

## Referee Comment (RC1) · Fryer (Referee) · 6 Dec 2018

This is a well-written straightforward paper which reaches the clear conclusion that many tsunamigenic landslides are seismically undetectable. This result is important. I recommend that the paper be published with only minor revision.

I have four concerns, which could be addressed by lengthening the paper only slightly. First, the paper treats all submarine landslides as if they are the same, but landslides have a broad range of characteristics which should at least be mentioned. The landslide types most important in generating tsunamis are slumps and debris avalanches. A slump is a landslide in which a coherent block of material slides downslope on a rota-

tional slip surface. The event is of relatively short duration (a few tens of seconds) and downslope motion is relatively small, so remote detection of the slope failure is going to be a challenge. The PNG tsunami came from such a source. A debris avalanche, by contrast, involves complete disintegration of the sliding body, motion lasts for a long time (conceivably several minutes), and both downslope motion and runout at the base of the slope are large. The St. Helens landslide the authors refer to was a debris avalanche. Even if we cannot warn of slump-generated tsunamis, the larger signals from debris avalanches should allow us to warn of those events.

Second, no mention is made of those landslides which have been detected remotely. Ekström and Stark (Science, March 2013), have identified large subaerial landslides from broadband seismology, while Caplan-Auerbach, et al. (GRL, May 2001) have detected submarine landslides from hydrophone data (note that in both these cases the landslides identified were debris avalanches rather than slumps). No mention of either of these is made in the paper.

Third, Katsumata, et al., credit Kodaira, et al. for the suggestion that a landslide supplemented the 2011 Tohoku tsunami, but make no mention of the more detailed analysis and modeling by Tappin, et al. (Marine Geology, 2014), which pretty much confirms that there was a landslide. There should at least be a reference to the paper of Tappin, et al.

Fourth, Katsumata, et al. ignore hydrophones in their discussion of potential detection systems and instead suggest direct detection of the tsunami via pressure gauges like S-net. But direct detection is intrinsically slow because you have to wait for the tsunami to reach your sensor. Since sound waves in the ocean travel faster than the tsunami, hydrophones potentially provide more warning time and would therefore be superior. Again, the Caplan-Auerbach paper is relevant here.

I have only one specific comment on the writing. On page 4, lines 9-14, in a rather awkward passage, the Mediterranean is described as "seismically inactive." I understand

the point that the authors are trying to make, but most readers will not. I recommend instead that they write something like "The Mediterranean is a region where seismic activity is low enough that most of the known tsunamis have been caused by landslides (Salamon, et al., 2011). Because of the greater seismicity, such conditions do not exist in southeast Asia. It is plausible there that heavy rainfall and rapid deposition of terrigenous sediment offshore might contribute to the occurrence of submarine landslides, including the PNG landslide, despite their location in a seismically active region."

---

## Referee Comment (RC2) · Anonymous Referee #2 · 15 Dec 2018

This is a paper on a very important issue to study if tsunamigenic submarine slides are seismically detectable, in the tens of minutes after a strong earthquake. The conclusion for this specific event , obtained with a set of seismic station located one at 150 km , a second at 900 km and all other at more than 2000 km (teleseismic distance), is negative. I consider that this paper needs major revision, for various reasons.

First, Katsumata et al are trying to find the signature of the submarine slide in the seismic record, without mentioning and describing in the figures that 4 aftershocks were identified during the 22 Minutes following the main shock (Synolakis 2002). In Figure 3 JAY record shows that waves of the two largest aftershocks are arriving at

09:09.30 and 09:10:30 (not mentioned by the authors). Synolakis specified that one of the aftershocks at 09:02 mb 4.4 could correspond to the submarine landslide. In Figure 1 an additional record filtered 0.1- 1 s would probably help to show the high frequency waves of this " aftershock", in fact the slide. Long duration of main chock (> 2-5 min) and aftershocks occurring in the tens of minutes after the main shock could definitively masked the waves generated by the submarine landslide generated in the 10-20 minutes following the quake. But as this event was identified and located by seismic waves picking and measurement, signal should be visible on the JAY record at higher frequency (> 1 Hz).

Second, the synthetic records obtained by modeling by Katsumata et al, for the closest station JAY, are of much bigger than the waves on records, in the 50-100s band. As mentioned by Fryer, the slide could be a slump type, or debris avalanche type, and in addition, the rheology parameters could vary extremely. The conclusion is, because no signal is visible in the bandwidth 50-100 s on the JAY station record, the hypothesis of the synthetic source and propagation performed by the authors is probably not correct. Katsumata et al should performed other synthetic records, knowing that in the Figure 2 shows that , at JAY station, in the bandwidth 0.5s to 100s, no clear signal is visible at the theoretical arrival time of the waves of the slide.

Third, other processing methods exist to help to identify waves visually or by signal processing : computation of spectrograms is one of the efficient method, and computation of polarization parameters of waves.

Conclusion : Katsumata et al finally demonstrate that the synthetic record obtained for JAY seismic station doesn't match with the observed record in the specific band (50-100s). JAY record shows that no signal is visible in the bandwidth of 0.5s to 100s, 13 minutes after the quake, when the slide waves are expected. The conclusion of the authors is not relevant : other type and parameters of the slide could be modeled to compute synthetic records and compare with JAY record in higher frequency band (0.1 - 1s). Detect, identify and warn a tsunami due to submarine or aerial slide following

large earthquake is definitively a complex challenge, essentially because of the duration of the quake and also the number and magnitude of aftershocks. As mentioned by Katsumata et al., S-net and DONET equipped with accelerometers, seismometers and pressure sensors are the most likely candidates to detect and warn submarine landslide. Nevertheless seismic arrays and seismic stations located closer to the slide (< 100 km) could be able to detect slide waves. In addition, hydroacoustic arrays (Synolakis) and coastal seismic station located on islands close to the epicenter could also help to detect T phase generated by the quakes and those generated by the slide. This paper needs major revision.

---

## Referee Comment (RC3) · Tappin (Referee) · 15 Dec 2018

Review nhess-2018-317 PNG Been through the paper and two previous reviews (RC1 and RC2), and agree with both. Fryer identifies missing references and the potential for T-phase warning which could provide an alternative approach. I question some of the interpretations of the different tsunami mechanisms in the Mediterranean. There are major seismic hazards here, such as Messina, 1908 and the EBTP of 365 AD. There are other earthquake tsunamis in the Ionian Sea. The work of Salamon et al., is questionable as it is based on the interpretation of ancient texts in earthquake identification (intensities) and not on modern methodologies of marine mapping and

seismological identification of earthquakes and their magnitudes. A more appropriate reference is: Papadopoulos, G.A., Gràcia, E., Urgeles, R., Sallares, V., De Martini, P.M., Pantosti, D., González, M., Yalciner, A.C., Mascle, J., Sakellariou, D., Salamon, A., Tinti, S., Karastathis, V., Fokaefs, A., Camerlenghi, A., Novikova, T., Papageorgiou, A., 2014. Historical and pre-historical tsunamis in the Mediterranean and its connected seas: Geological signatures, generation mechanisms and coastal impacts. Marine Geology 354, 81-109.

The anonymous reviewer (RC2) undoubtedly identifies a fundamental flaw in the analysis. The PNG 09.02 'seismic' event is unique and was identified by Synolakis et al. as reflecting the slump movement, not a seismic event. The modelled and observed signals do not match. The authors have to go back and revisit the frequency signal at this time, and address this before the paper can be published – it is fundamental to the papers conclusions and the possibilities for tsunami warning from submarine landslides.

To confirm, the PNG landslide was a slump. I mapped it.

The Salamon reference is missing. I don't think there is a peer reviewed paper from 2011. 2007 maybe from the eastern Med?

David R Tappin
* * *

---

## Author Comment (AC1) · 7 Jan 2019

Dear Referee #1 (Prof. Fryer)

We changed the manuscript according to the commends from reviewers.

First, the paper treats all submarine landslides as if they are the same, but landslides have a broad range of characteristics which should at least be mentioned. The landslide types most important in generating tsunamis are slumps and debris avalanches. A slump is a landslide in which a coherent block of material slides downslope on a rotational slip surface. The event is of relatively short duration (a

few tens of seconds) and downslope motion is relatively small, so remote detection of the slope failure is going to be a challenge. The PNG tsunami came from such a source. A debris avalanche, by contrast, involves complete disintegration of the sliding body, motion lasts for a long time (conceivably several minutes), and both downslope motion and runout at the base of the slope are large. The St. Helens landslide the authors refer to was a debris avalanche. Even if we cannot warn of slump-generated tsunamis, the larger signals from debris avalanches should allow us to warn of those events.

- We added description about various types of submarine mass failure referring to Schwab et al. (1993), and mentioned that the 1998 PNG event had a relatively short travel distance.

  It is considered that the amplitudes of seismic waves are proportional to the peak force. Even if the duration of mass motion is long, the amplitude of the seismic waves would not be different for the same size of forces. We added this description in the section of synthetic seismogram.

Second, no mention is made of those landslides which have been detected remotely. Ekström and Stark (Science, March 2013), have identified large subaerial landslides from broadband seismology, while Caplan-Auerbach, et al. (GRL, May 2001) have detected submarine landslides from hydrophone data (note that in both these cases the landslides identified were debris avalanches rather than slumps). No mention of either of these is made in the paper.

- Landslides detected remotely by seismic waves are mention in the section of "Discussion". However, Ekström and Stark (2013) was not referred to. We added descriptions about Ekström and Stark (2013) in the the "Discussion" section with other references about seismic waves from landslides.

  As to detection landslides with hydrophones, Synolakis et al. (2002) showed a hydrophone record from the 1998 PNG landslide. We added Caplan-

Auerbach, et al. (2001) as a reference. It is considered that the detection of waves from landslides are possible with hydrophones. However, we consider that identification of landslide and estimation of size of landslides are difficult with hydrophone data. We added such description in "Introduction".

Third, Katsumata, et al., credit Kodaira, et al. for the suggestion that a landslide supplemented the 2011 Tohoku tsunami, but make no mention of the more detailed analysis and modeling by Tappin, et al. (Marine Geology, 2014), which pretty much confirms that there was a landslide. There should at least be a reference to the paper of Tappin, et al.

- We changed the manuscript with adding Tappin, et al. (2014) as a reference.

Fourth, Katsumata, et al. ignore hydrophones in their discussion of potential detection systems and instead suggest direct detection of the tsunami via pressure gauges like S-net. But direct detection is intrinsically slow because you have to wait for the tsunami to reach your sensor. Since sound waves in the ocean travel faster than the tsunami, hydrophones potentially provide more warning time and would therefore be superior. Again, the Caplan-Auerbach paper is relevant here.

- We considered that size estimation is indispensable for tsunami warning purpose. Passband of instruments should cover the process duration to estimate the size of the event properly. We added description in "Introduction" to claim that hydrophone is not useful for estimation of landslide size.

I have only one specific comment on the writing. On page 4, lines 9-14, in a rather awkward passage, the Mediterranean is described as "seismically inactive." I understand the point that the authors are trying to make, but most readers will not. I recommend instead that they write something like "The Mediterranean is a region where seismic activity is low enough that most of the known tsunamis have been

caused by landslides (Salamon, et al., 2011). Because of the greater seismicity, such conditions do not exist in southeast Asia. It is plausible there that heavy rainfall and rapid deposition of terrigenous sediment offshore might contribute to the occurrence of submarine landslides, including the PNG landslide, despite their location in a seismically active region."

- We followed this kind suggestion.

Please also note the supplement to this comment:
https://www.nat-hazards-earth-syst-sci-discuss.net/nhess-2018-317/nhess-2018-317-AC1-supplement.pdf

**Supplement:**

**Detectability of seismic waves from the submarine landslide that caused the 1998 Papua New Guinea tsunami**

Akio Katsumata[1], Yasuhiro Yoshida[2], Kenji Nakata[1], Kenichi Fujita[1], Masayuki Tanaka[1], Koji Tamaribuchi[1], Takahito Nishimiya[1], and Akio Kobayashi[1]

[1]Meteorological Research Institute, Japan Meteorological Agency, 1-1 Nagamine, Tsukuba, Ibaraki Prefecture, Japan
[2]Meteorological College, Japan Meteorological Agency, 7-4-81 Asahi-cho, Kashiwa, Chiba Prefecture, Japan

**Correspondence:** Akio Katsumata (akatsuma@mri-jma.go.jp)

**Abstract.** On 17 July 1998, a tsunami caused serious damage on the northern coast of Papua New Guinea about 20 min after the mainshock of an $M_w$ 7.0 earthquake. The tsunami has been attributed to a submarine landslide that occurred about 13 min after the mainshock because its arrival at the coast was too late and its height too great to be the direct result of the fault slip of the earthquake. Bathymetric data recorded after the tsunami revealed an amphitheater-like structure that was consistent with a recent submarine landslide. Most current tsunami warning systems are based on analysis of the early arrivals of seismic waves generated by an earthquake. In this study we investigated whether evidence of the landslide could be identified in the coda waves recorded after the mainshock. Based on previous studies of the tsunami source, we constructed synthetic seismograms to represent the submarine landslide and compared them to the observed coda waves of the preceding earthquake, with particular attention to the period around 13 min after the mainshock in frequency ranges close to the landslide duration. We found phases possibly corresponding to the landslide event. However, they were easily covered with coda waves from the mainshock. We concluded that the 1998 landslide was too small to be evident in the coda waves following the magnitude 7 earthquake.

*Copyright statement.* TEXT

**1 Introduction**

A tsunami struck the north coast of Papua New Guinea (PNG) on 17 July 1998. The height of the tsunami was more than 10 m and its passage resulted in the deaths of more than two thousand people (e.g., Kawata et al. , 1999; Tappin et al. , 1999; Synolakis et al. , 2002). The tsunami struck the coast about 20 min after an earthquake (Kawata et al. , 1999) of $M_w$ 7.0 (GCMT, Dziewonski et al. , 1981; Ekström et al. , 2012). The arrival time of the tsunami was too late and its height too great for it to have been a direct result of the fault slip of the earthquake. Moreover, bathymetric data recorded after the earthquake revealed an amphitheater-like structure consistent with recent submarine slumping about 20 km off the coast (Tappin et al. , 1999). It was therefore presumed that the tsunami was caused by a submarine landslide (e.g., Tappin et al. , 1999; Heinrich et al. , 2000; Synolakis et al. , 2002). Numerical simulations of the landslide (e.g., Tappin et al. , 1999; Watts et al. ,

2003; Lynett et al. , 2003) provided landslide masses of 6.4 km$^3$ (Tappin et al. , 2008), 4 km$^3$ (Heinrich et al. , 2000), and 9 km$^3$ (Watts et al. , 2003).

Various types of submarine mass failures have been recognized, and some mass flows traveled hudreds of kilometers (e.g., Schwab et al. , 1993). The mass motion distance of the 1998 PNG event was estimated at a relatively short one (e.g., 766m by Watts et al. , 2005; 980m by Tappin et al. , 2008), and was a slump type.

Early warnings of impending tsunamis are usually issued based on analyses of the early arrivals of seismic waves generated by earthquakes. If the submarine landslide after the 1998 PNG earthquake had been detected in the seismic data recorded immediately after the earthquake, it might have been possible to provide advanced warnings to PNG residents of the imminent tsunami. In this study, we investigated whether the seismic signature of the submarine landslide was present in the coda waves of the seismic data recorded after the mainshock of the 1998 earthquake. It is necessary to estimate size of a landslide as well as its source type for tsunami warning purpose. The 1998 PNG events was identified in a hydrophone record (Synolakis et al. , 2002; Okal , 2003). Caplan-Auerbach et al. (2001) reported hydrophone records from submarine landslides at Kilauea volcano. When source size is estimated from observed amplitues, frequency range of records should cover the process duration for a proper estimattion (e.g., Aki , 1967). The duration of the 1998 PNG landslide was estimated at several tens to 100 s (Watts et al. , 2003; Tappin et al. , 2008). The passband of the hydrophone arrays (e.g., 1–40 Hz of the PMEL hydrophone array, Caplan-Auerbach et al. , 2001) seems to be insufficient to estimate the size of the landslide. We focus on seismic records of frequency ranges of 0.01–1 Hz.

**2   Observed seismic records**

[revised manuscript text omitted]

Ekström et al. (2012) presented scaling relationship of landslides. The force of $7 \times 10^{12}$ N correspods to 5.8 of surface wave magnitude according to the scaling relation. A vertical displacement of 0.05 mm is expected at a epicentral distance of 2000 km and at a period of 20 s for this surface wave magnitude ( Vaněk et al., 1962). The amplitude of 0.05 mm is comparable to

observed amplitudes of coda waves at stations of epicentral distance around 2000km shown in 3 for passband around 20 s. It is unlikely to recognize the occurrence of the landslide based on the seismic records after the large earthquake.

**4 Discussion**

[revised manuscript text omitted]

Takeo, M.: Near-field synthetic seismograms taking into account the effects of anelasticity -The effects of anelastic attenuation on seismo-grams caused by a sedimentary layer-, Pap. Meteorol. Geophys., 36, 245–257, 1985, doi:10.2467/mripapers.36.245. (In Japanese with English abstract)

Tappin, D. R., Matsumoto, T., Watts, P., Satake, K., McMurtry, G. M., Matsuyama, M., Lafoy, Y., Tsuji, Y., Kanamatsu, T., Lus, W., Iwabuchi, Y., Yeh, H., Matsumotu, Y., Nakamura, M., Mahoi, M., Hill, P., Crook, K., Anton, L., and Walsh, J. P.: Sediment slump likely caused 1998 Papua New Guinea tsunami, Eos, Trans. Am. Geophys. Union, 80(30), 329–344, 1999.

Tappin, D. R. , Watts, P., Grilli, S. T.: The Papua New Guinea tsunami of 17 July 1998: anatomy of a catastrophic event, Nat. Hazard Earth Sys., 8, 243–266, doi:10.5194/nhess-8-243-2008, 2008

Tappin, D. R., Grilli, S. T., Harris, J. C., Geller, R. J., Masterlark, T., Kirby, J. T., Shi, F., Ma, G., Thingbaijam, , K.K.S., and Mai, P. M.: Did a submarine landslide contribute to the 2011 Tohoku tsunami?, MAR. GEOL., 357, 344–361, doi:10.1016/j.margeo.2014.09.043, 2014.

Uehira, K., Kanazawa, T., Mochizuki, M., Fujimoto, H., Noguchi, S., Shinbo, T. Shiomi, K., Kunugi, T., Aoi, S., Matsumoto, T., Sekiguchi, S., Okada, Y., Shinohara, M., and Yamada, T.: Outline of Seafloor Observation Network for Earthquakes and Tsunamis along the Japan Trench (S-net), EGU General Assembly, Vienna, Austria, 17–22 April 2016, EPSC2016-13832, 2016.

Vaněk, J., Zatopek, A., Karnik, V., Kondroskaya, N. V., Riznichenko, Y. V., Savarensky, E. F., Solov'ev, S. L., and Shebalin, N.V.: Standardization of magnitude scales, Izu. Acak. Nauk. SSSR, Ser. Geofiz., 2, 108-111, 1962.

Watts, P., Grilli, S. T., Kirby, J. T., Fryer, G. J., and Tappin, D. R.: Landslide tsunami case studies using a Boussinesq model and a fully nonlinear tsunami generation model, Nat. Hazard Earth Sys., 3, 391–402, doi:10.5194/nhess-3-391-2003, 2003.

Watts, P., Grill, S. T., M.ASCE, Tappin, D. R., and Fryer, G. J.: Tsunami generation by submarine mass failure II: Predictive equations and case studies, J. Waterw. Port C-ASCE, 131, 298–310, doi:10.1061/(ASCE)0733-950X(2005)131:6(298), 2005.

[revised manuscript text omitted]

---

## Author Comment (AC3) · 7 Jan 2019

Dear Prof. Tappin,

We are grateful to giving comments on our manuscript. We changed the manuscript according to your suggestion.

Fryer identifies missing references and the potential for T-phase warning which could provide an alternative approach.

- We considered that size estimation is indispensable for tsunami warning

purpose. Passband of instruments should cover the process duration to estimate the size of the event properly. We added description in "Introduction" to claim that hydrophone is not useful for estimation of landslide size.

I question some of the interpretations of the different tsunami mechanisms in the Mediterranean. There are major seismic hazards here, such as Messina, 1908 and the EBTP of 365 AD. There are other earthquake tsunamis in the Ionian Sea. The work of Salamon et al., is questionable as it is based on the interpretation of ancient texts in earthquake identification (intensities) and not on modern methodologies of marine mapping and seismological identification of earthquakes and their magnitudes. A more appropriate reference is: Papadopoulos, G.A., Gárcia, E., Urgeles, R., Sallares, V., De Martini, P.M., Pantosti, D., González, M., Yalciner, A.C., Mascle, J., Sakellariou, D., Salamon, A., Tinti, S., Karastathis, V., Fokaefs, A., Camerlenghi, A., Novikova, T., Papageorgiou, A., 2014. Historical and pre-historical tsunamis in the Mediterranean and its connected seas: Geological signatures, generation mechanisms and coastal impacts. Marine Geology 354, 81-109.

- We added this suggested paper. However sources of many tsunamis were not identified in this paper. We did not change the context of "Discussion" so much.

The anonymous reviewer (RC2) undoubtedly identifies a fundamental flaw in the analysis. The PNG 09.02 'seismic' event is unique and was identified by Synolakis et al. as reflecting the slump movement, not a seismic event. The modelled and observed signals do not match. The authors have to go back and revisit the frequency signal at this time, and address this before the paper can be published – it is fundamental to the papers conclusions and the possibilities for tsunami warning from submarine landslides.

- We are interested in seismic signals which have frequency band close to the landslide duration to estimate size of tsunami height. We added description in "Introduction" to make it clear.

To confirm, the PNG landslide was a slump. I mapped it.

- We added your paper (2008) to refer to the travel distance of the 1998 PNG event.

The Salamon reference is missing. I don't think there is a peer reviewed paper from 2011. 2007 maybe from the eastern Med?

- We received this paper PDF directly from the author. We couldn't reach the journal page, either. So we replaced this reference to BSSA, 2007.

Please also note the supplement to this comment:
https://www.nat-hazards-earth-syst-sci-discuss.net/nhess-2018-317/nhess-2018-317-AC3-supplement.pdf

**Supplement:**

**Detectability of seismic waves from the submarine landslide that caused the 1998 Papua New Guinea tsunami**

Akio Katsumata[1], Yasuhiro Yoshida[2], Kenji Nakata[1], Kenichi Fujita[1], Masayuki Tanaka[1], Koji Tamaribuchi[1], Takahito Nishimiya[1], and Akio Kobayashi[1]

[1]Meteorological Research Institute, Japan Meteorological Agency, 1-1 Nagamine, Tsukuba, Ibaraki Prefecture, Japan
[2]Meteorological College, Japan Meteorological Agency, 7-4-81 Asahi-cho, Kashiwa, Chiba Prefecture, Japan

**Correspondence:** Akio Katsumata (akatsuma@mri-jma.go.jp)

**Abstract.** On 17 July 1998, a tsunami caused serious damage on the northern coast of Papua New Guinea about 20 min after the mainshock of an $M_w$ 7.0 earthquake. The tsunami has been attributed to a submarine landslide that occurred about 13 min after the mainshock because its arrival at the coast was too late and its height too great to be the direct result of the fault slip of the earthquake. Bathymetric data recorded after the tsunami revealed an amphitheater-like structure that was consistent with a recent submarine landslide. Most current tsunami warning systems are based on analysis of the early arrivals of seismic waves generated by an earthquake. In this study we investigated whether evidence of the landslide could be identified in the coda waves recorded after the mainshock. Based on previous studies of the tsunami source, we constructed synthetic seismograms to represent the submarine landslide and compared them to the observed coda waves of the preceding earthquake, with particular attention to the period around 13 min after the mainshock in frequency ranges close to the landslide duration. We found phases possibly corresponding to the landslide event. However, they were easily covered with coda waves from the mainshock. We concluded that the 1998 landslide was too small to be evident in the coda waves following the magnitude 7 earthquake.

*Copyright statement.* TEXT

**1 Introduction**

A tsunami struck the north coast of Papua New Guinea (PNG) on 17 July 1998. The height of the tsunami was more than 10 m and its passage resulted in the deaths of more than two thousand people (e.g., Kawata et al. , 1999; Tappin et al. , 1999; Synolakis et al. , 2002). The tsunami struck the coast about 20 min after an earthquake (Kawata et al. , 1999) of $M_w$ 7.0 (GCMT, Dziewonski et al. , 1981; Ekström et al. , 2012). The arrival time of the tsunami was too late and its height too great for it to have been a direct result of the fault slip of the earthquake. Moreover, bathymetric data recorded after the earthquake revealed an amphitheater-like structure consistent with recent submarine slumping about 20 km off the coast (Tappin et al. , 1999). It was therefore presumed that the tsunami was caused by a submarine landslide (e.g., Tappin et al. , 1999; Heinrich et al. , 2000; Synolakis et al. , 2002). Numerical simulations of the landslide (e.g., Tappin et al. , 1999; Watts et al. ,

2003; Lynett et al. , 2003) provided landslide masses of 6.4 km$^3$ (Tappin et al. , 2008), 4 km$^3$ (Heinrich et al. , 2000), and 9 km$^3$ (Watts et al. , 2003).

Various types of submarine mass failures have been recognized, and some mass flows traveled hudreds of kilometers (e.g., Schwab et al. , 1993). The mass motion distance of the 1998 PNG event was estimated at a relatively short one (e.g., 766m by Watts et al. , 2005; 980m by Tappin et al. , 2008), and was a slump type.

Early warnings of impending tsunamis are usually issued based on analyses of the early arrivals of seismic waves generated by earthquakes. If the submarine landslide after the 1998 PNG earthquake had been detected in the seismic data recorded immediately after the earthquake, it might have been possible to provide advanced warnings to PNG residents of the imminent tsunami. In this study, we investigated whether the seismic signature of the submarine landslide was present in the coda waves of the seismic data recorded after the mainshock of the 1998 earthquake. It is necessary to estimate size of a landslide as well as its source type for tsunami warning purpose. The 1998 PNG events was identified in a hydrophone record (Synolakis et al. , 2002; Okal , 2003). Caplan-Auerbach et al. (2001) reported hydrophone records from submarine landslides at Kilauea volcano. When source size is estimated from observed amplitues, frequency range of records should cover the process duration for a proper estimattion (e.g., Aki , 1967). The duration of the 1998 PNG landslide was estimated at several tens to 100 s (Watts et al. , 2003; Tappin et al. , 2008). The passband of the hydrophone arrays (e.g., 1–40 Hz of the PMEL hydrophone array, Caplan-Auerbach et al. , 2001) seems to be insufficient to estimate the size of the landslide. We focus on seismic records of frequency ranges of 0.01–1 Hz.

**2   Observed seismic records**

[revised manuscript text omitted]

Ekström et al. (2012) presented scaling relationship of landslides. The force of $7 \times 10^{12}$ N correspods to 5.8 of surface wave magnitude according to the scaling relation. A vertical displacement of 0.05 mm is expected at a epicentral distance of 2000 km and at a period of 20 s for this surface wave magnitude ( Vaněk et al., 1962). The amplitude of 0.05 mm is comparable to

observed amplitudes of coda waves at stations of epicentral distance around 2000km shown in 3 for passband around 20 s. It is unlikely to recognize the occurrence of the landslide based on the seismic records after the large earthquake.

**4 Discussion**

[revised manuscript text omitted]

Takeo, M.: Near-field synthetic seismograms taking into account the effects of anelasticity -The effects of anelastic attenuation on seismo-grams caused by a sedimentary layer-, Pap. Meteorol. Geophys., 36, 245–257, 1985, doi:10.2467/mripapers.36.245. (In Japanese with English abstract)

Tappin, D. R., Matsumoto, T., Watts, P., Satake, K., McMurtry, G. M., Matsuyama, M., Lafoy, Y., Tsuji, Y., Kanamatsu, T., Lus, W., Iwabuchi, Y., Yeh, H., Matsumotu, Y., Nakamura, M., Mahoi, M., Hill, P., Crook, K., Anton, L., and Walsh, J. P.: Sediment slump likely caused 1998 Papua New Guinea tsunami, Eos, Trans. Am. Geophys. Union, 80(30), 329–344, 1999.

Tappin, D. R. , Watts, P., Grilli, S. T.: The Papua New Guinea tsunami of 17 July 1998: anatomy of a catastrophic event, Nat. Hazard Earth Sys., 8, 243–266, doi:10.5194/nhess-8-243-2008, 2008

Tappin, D. R., Grilli, S. T., Harris, J. C., Geller, R. J., Masterlark, T., Kirby, J. T., Shi, F., Ma, G., Thingbaijam, , K.K.S., and Mai, P. M.: Did a submarine landslide contribute to the 2011 Tohoku tsunami?, MAR. GEOL., 357, 344–361, doi:10.1016/j.margeo.2014.09.043, 2014.

Uehira, K., Kanazawa, T., Mochizuki, M., Fujimoto, H., Noguchi, S., Shinbo, T. Shiomi, K., Kunugi, T., Aoi, S., Matsumoto, T., Sekiguchi, S., Okada, Y., Shinohara, M., and Yamada, T.: Outline of Seafloor Observation Network for Earthquakes and Tsunamis along the Japan Trench (S-net), EGU General Assembly, Vienna, Austria, 17–22 April 2016, EPSC2016-13832, 2016.

Vaněk, J., Zatopek, A., Karnik, V., Kondroskaya, N. V., Riznichenko, Y. V., Savarensky, E. F., Solov'ev, S. L., and Shebalin, N.V.: Standardization of magnitude scales, Izu. Acak. Nauk. SSSR, Ser. Geofiz., 2, 108-111, 1962.

Watts, P., Grilli, S. T., Kirby, J. T., Fryer, G. J., and Tappin, D. R.: Landslide tsunami case studies using a Boussinesq model and a fully nonlinear tsunami generation model, Nat. Hazard Earth Sys., 3, 391–402, doi:10.5194/nhess-3-391-2003, 2003.

Watts, P., Grill, S. T., M.ASCE, Tappin, D. R., and Fryer, G. J.: Tsunami generation by submarine mass failure II: Predictive equations and case studies, J. Waterw. Port C-ASCE, 131, 298–310, doi:10.1061/(ASCE)0733-950X(2005)131:6(298), 2005.

[revised manuscript text omitted]

---

## Short Comment (SC1) · 8 Jan 2019

Dear reviewer #2,

I show high-pass filtered records ( > 1 Hz) at JAY station. This frequency range is not of our interest. However I shall upload these records to show the aftershocks can be recognized also at JAY station.

**Fig. 1.** High-pass filtered records at JAY

---

## Short Comment (SC2) · 13 Jan 2019

We are grateful to giving comments on our manuscript. We modified the manuscript (supplement) accoring to your comments.

The main argument of the paper is that there is no detectable seismic signature to the landslide which generated the catastrophic PNG tsunami of 17 July 1998. This statement directly contradicts the work of Okal [2003], in which I presented (on Figure 3) and discussed in detail the record of the landslide at the same station JAY allegedly studied by the authors. It is clear that they used the wrong (very

low-frequency) filters, and thus missed the signal. They do not justify working in such inadequate frequency bands, and completely ignore the detailed analysis of seismic and hydroacoustic phases which went into my 2003 paper.

- Our main concern is tsunami warning. To issue a proper tsunami warning, height of tsunami should be estimated. Size of landslide is an indispens-able factor to estimate tsunami height. Detection of a landslide may be possible with short-period seismometers or hydrophones. However those instruments are not useful to estimate size of the total mass. The duration of the waves would reflect the duration of of the landslide. However it is not directly connected to the mass. We missed to describe these matters in the submitted manuscript. We changed the manuscript so that out interest on long period seismic records would be expressed explicitly.

I note on Page 4, Line 9 the statement "The Mediterranean is a seismically inactive region"! This is completely false. The USGS catalog contains 1132 events with at least one magnitude reaching 5 or greater for the period 1963–2015, between latitudes 30 and 45°N, longitudes -5 and 35°E, and depths 0 and 100 km... This factually wrong scientific statement takes an insulting societal tone when confronted to the memory of the thousands of victims of earthquakes in the Mediterranean Basin, documented since historic times.

- We must admit that the expression of "inactive" was improper. We changed expression following Prof. Fryer.

It is wrong to use the reference to Tappin et al. [2008] to suggest that the slide underwent a "deceleration stage affected by interaction of the sliding mass with sea water". All submarine slides will feature such interaction. What was unique in the PNG slide was that it was stopped abruptly when it abutted against the opposite

wall of the amphitheater in which it took place. All of this was explained in detail by Synolakis et al. [2002] and Okal [2003]; as mentioned above, the authors seem to ignore the latter paper, as they ignore the fundamental paper by Sweet and Silver [2003], who conducted the in situ discovery and study of the slide.

- We did not refer to Tappin et al. [2008] to explain the acceleration and deceleration stages. We referred to Tappin et al. [2008] as the estimation of the landslide duration. We added Ekström and Stark (2013) as the acceleration and deceleration stages. We also added Synolakis et al. [2002] and Okal [2003] to mention the short duration estimated from hydrophone data.

The dynamics of the underwater PNG landslide and of the Mt. St. Helens one are totally different, given that the latter was caused by an atmospheric explosion, and reached velocities of 70 m/s (as documented from films) which cannot be sustained by underwater landslides.

- We referred to Kanamori and Given (1982) just to compare the force values. We do not discuss the difference in the sliding process.

The proposal to densely instrument the seafloor in order to detect and identify in real time a landslide and issue a warning is naive in the context of the PNG tsunami, given that the whole process would have to be realized in a few minutes. Most of the casualties at Sissano resulted from the lack of an escape route: the residents were trapped on a narrow spit of land between the Bismarck Sea and Sissano Lagoon. The only survivors had managed to climb the few trees which were not uprooted. As such distances, the only reliable means of tsunami mitigation is proper planning (the village should not have been built on the spit), and in real-time, self-evacuation. References Okal, E.A., T waves from the 1998 Papua New Guinea earthquake and its Fryer identifies missing references and the potential for T-phase warning which could provide an alternative approach.

- Evacuation method is a very important factor for mitigation of tsunami disaster. We consider that awareness of coming tsunami is also an important one at the same time. Only when all factors are controlled properly, the victims would be reduced. So we think we should pay attention to awareness of coming tsunami.

Please also note the supplement to this comment:
https://www.nat-hazards-earth-syst-sci-discuss.net/nhess-2018-317/nhess-2018-317-SC2-supplement.pdf

**Supplement:**

**Detectability of seismic waves from the submarine landslide that caused the 1998 Papua New Guinea tsunami**

Akio Katsumata[1], Yasuhiro Yoshida[2], Kenji Nakata[1], Kenichi Fujita[1], Masayuki Tanaka[1], Koji Tamaribuchi[1], Takahito Nishimiya[1], and Akio Kobayashi[1]

[1]Meteorological Research Institute, Japan Meteorological Agency, 1-1 Nagamine, Tsukuba, Ibaraki Prefecture, Japan
[2]Meteorological College, Japan Meteorological Agency, 7-4-81 Asahi-cho, Kashiwa, Chiba Prefecture, Japan

**Correspondence:** Akio Katsumata (akatsuma@mri-jma.go.jp)

**Abstract.** On 17 July 1998, a tsunami caused serious damage on the northern coast of Papua New Guinea about 20 min after the mainshock of an $M_w$ 7.0 earthquake. The tsunami has been attributed to a submarine landslide that occurred about 13 min after the mainshock because its arrival at the coast was too late and its height too great to be the direct result of the fault slip of the earthquake. Bathymetric data recorded after the tsunami revealed an amphitheater-like structure that was consistent with a recent submarine landslide. Most current tsunami warning systems are based on analysis of the early arrivals of seismic waves generated by an earthquake. In this study we investigated whether evidence of the landslide could be identified in the coda waves recorded after the mainshock. Based on previous studies of the tsunami source, we constructed synthetic seismograms to represent the submarine landslide and compared them to the observed coda waves of the preceding earthquake, with particular attention to the period around 13 min after the mainshock in frequency ranges close to the landslide duration. We found phases possibly corresponding to the landslide event. However, they were easily covered with coda waves from the mainshock. We concluded that the 1998 landslide was too small to be evident in the coda waves following the magnitude 7 earthquake.

*Copyright statement.* TEXT

**1 Introduction**

A tsunami struck the north coast of Papua New Guinea (PNG) on 17 July 1998. The height of the tsunami was more than 10 m and its passage resulted in the deaths of more than two thousand people (e.g., Kawata et al. , 1999; Tappin et al. , 1999; Synolakis et al. , 2002). The tsunami struck the coast about 20 min after an earthquake (Kawata et al. , 1999) of $M_w$ 7.0 (GCMT, Dziewonski et al. , 1981; Ekström et al. , 2012). The arrival time of the tsunami was too late and its height too great for it to have been a direct result of the fault slip of the earthquake. Moreover, bathymetric data recorded after the earthquake revealed an amphitheater-like structure consistent with recent submarine slumping about 20 km off the coast (Tappin et al. , 1999). It was therefore presumed that the tsunami was caused by a submarine landslide (e.g., Tappin et al. , 1999; Heinrich et al. , 2000; Synolakis et al. , 2002). Numerical simulations of the landslide (e.g., Tappin et al. , 1999; Watts et al. ,

2003; Lynett et al. , 2003) provided landslide masses of 6.4 km$^3$ (Tappin et al. , 2008), 4 km$^3$ (Heinrich et al. , 2000), and 9 km$^3$ (Watts et al. , 2003).

Various types of submarine mass failures have been recognized, and some mass flows traveled hundreds of kilometers (e.g., Schwab et al. , 1993). The mass motion distance of the 1998 PNG event was estimated at a relatively short one (e.g., 766 m by Watts et al. , 2005; 980 m by Tappin et al. , 2008), and was a slump type.

Early warnings of impending tsunamis are usually issued based on analyses of the early arrivals of seismic waves generated by earthquakes. If the submarine landslide after the 1998 PNG earthquake had been detected in the seismic data recorded immediately after the earthquake, it might have been possible to provide advanced warnings to PNG residents of the imminent tsunami. In this study, we investigated whether the seismic signature of the submarine landslide was present in the coda waves of the seismic data recorded after the mainshock of the 1998 earthquake. It is necessary to estimate size of a landslide as well as its source type for tsunami warning purpose. The 1998 PNG events was identified in a hydrophone record (Synolakis et al. , 2002; Okal , 2003). Caplan-Auerbach et al. (2001) reported hydrophone records from submarine landslides at Kilauea volcano. When source size is estimated from observed amplitudes, frequency range of records should cover the process duration for a proper estismation (e.g., Aki , 1967). The duration of the 1998 PNG landslide was estimated at several tens to 100 s (Watts et al. , 2003; Tappin et al. , 2008). The passband of the hydrophone arrays (e.g., 1–40 Hz of the PMEL hydrophone array, Caplan-Auerbach et al. , 2001; 1–30 Hz at Wake Island, Okal , 2003) seems to be insufficient to estimate the size of the landslide. Whereas the duration of the waves would have reflected the duratoin of landslide (Synolakis et al. , 2002; Okal , 2003), the mass size and the duration are not directly related to each other. We focus on seismic records of frequency ranges of 0.01–1 Hz.

**2   Observed seismic records**

The broadband seismic data recorded during the $M_w$ 7.0 earthquake that preceded the 1998 tsunami has been archived at both the Earthquake Research Institute (ERI; The University of Tokyo) and the Incorporated Research Institutions for Seismology (IRIS; headquarters in Washington DC). The two seismic stations from which data were mainly used in this study were 146 km (Jayapura station, JAY) and 918 km (Port Moresby station, PMG) from the epicenter (Fig. 1).

Synolakis et al. (2002) calculated that the submarine landslide occurred 13 min after the mainshock on the basis of hydrophone data recorded at Wake Island. Heidarzadeh and Stake (2015) used tide gauge data to estimate that the landslide occurred 12–17 min after the mainshock. Synolakis et al. (2002) pointed out that seismic signal corresponding to the landslide could not be recognized at PMG. Okal (2003) showed a highpass-filtered seismic record (> 1 Hz) obtained at JAY station, and attributed an event at 09:02 to the landslide. Records at JAY show no significant events at around 13 min after the mainshock in the four frequency bands considered and spectrographs of 0.01-1 Hz (Fig. 2) . Aftershocks occurred at 09:02:06 ($m_b$, 4.4) and 09:06:03 (no magnitude reported). The event at 09:02:06 was considered to be directly related to the submarine landslide (Synolakis et al. , 2002; Okal , 2003). Those earthquakes are not evident in this frequency and amplitude range. Whereas any significant events are not seen in the records at six stations around source area (Fig. 3), small phases are recognized about 13

min after the S-arrival in the 50–100 s band of horizontal components at JAY and PMG (bold parts of seismic waves in Fig. 3). The particle motions of the bold parts were oriented to north-south at JAY and northeast-southwest at PMG, which were roughly consistent with those of the Love waves from the source area. However, there were not such events in the records at the other stations (Fig. 3).

**5  3  Comparison of synthetic seismograms representing the 1998 submarine landslide with coda waves from the 1998 earthquake**

We estimated the amplitudes of the seismic waves generated by the 1998 landslide on the basis of the work of Watts et al. (2003), as described below. Relatively large amplitudes were expected based on the model by Watts et al. (2003) because of its relatively large assumed mass.

10  When a landslide starts, the change in the force that has supported the landslide mass generates seismic waves. The force ($F$) generated by a landslide is defined by $F = ma$, where $m$ is the mass of the landslide and $a$ is its acceleration. Watts et al. (2003) assumed the landslide mass to be a half ellipsoid and, on the basis of the parameters given in Table 1, estimated its volume to be 9 km$^3$ ($\frac{\pi wbT}{6}$) and its mass to be $19 \times 10^{12}$ kg. Using this mass and the initial acceleration $a_0$ (0.36 m s$^{-2}$; Table 1) of Watts et al. (2003) gives an estimated force of $7 \times 10^{12}$ N. The volume determined by Watts et al. (2003) (9 km$^3$) is

15  larger than those of Heinrich et al. (2000) (4 km$^3$), Synolakis et al. (2002) (4 km$^3$), and Tappin et al. (2008) (6.4 km$^3$), and the acceleration of 0.36 m s$^{-2}$ is smaller than that used by Tappin et al. (2008) (0.47 m s$^{-2}$). The force and volume estimated here for the 1998 PNG landslide are comparable to those estimated by Kanamori and Given  (1982) for the landslide associated with the 1980 Mount St. Helens eruption ($10^{13}$ N and 2.5 km$^3$, respectively).

Amplitudes of far-field seismic waves are theoretically proportional to the amplitude of the acting force (e.g., Aki and Richards ,

20  2002). For a case of a longer-duration mass failure, the total energy would become greater. Even for such a case, the amplitudes of seismic waves should be constrained by the peak of the force.

The time history of a landslide must be assumed in order to estimate the amplitude of the resultant seismic wave. Watts et al. (2003) assumed a characteristic time of 32 s. Tappin et al. (2008) estimated that the sliding process lasted about 100 s. We assumed that the sliding process was completed within several tens of seconds with an initial acceleration stage followed by a

25  deceleration stage (Ekström et al. , 2012) as shown in Fig. 4), and that the total impulse would not have been balanced because part of the deceleration stage would be affected by interaction of the sliding mass with sea water. Here we assume the same peak values of acceleration for starting and stopping. Since it was estimated that the landslide occurred inside the existing amphitheatre (Synolakis et al. , 2002; Okal , 2003; Sweet and Silver , 2003), it may have stopped with a large acceleration. The curves for each of the four intervals defined on Fig. 4 were expressed as trigonometric functions (Table 2). Five different

[revised manuscript text omitted]

seismic activity is low enough that many of the known tsunamis may have been caused by  landslides (Salamon et al. , 2007; Papadopoulos et al. , 2014). Because of the greater seismicity, such conditions do not exist in southeast Asia. It is plausible  there that heavy rainfall and rapid deposition of terrigenous sediment offshore might contribute to the occurrence of submarine landslides, including the 1998 PNG landslide, despite their location in a seismically active region. A few tsunamis in Japan, where tectonic activity is very high, have also been attributed to submarine landslides (e.g., Baba et al. , 2012; Baba et al. , 2017).

Chao et al. (2018) considered the use of seismic records to detect landslides in order to provide early warnings of impending tsunamis. However, as we have demonstrated here, the seismic signature of large submarine landslides can be overwhelmed by the seismic coda waves generated by earthquakes, so it can be difficult to detect submarine landslides soon after large earthquakes by this method. An alternative method of providing early warnings for tsunamis generated by submarine landslides is needed. Monitoring by networks of ocean-bottom tsunami gauges such as those of DONET (Kawaguchi et al. , 2015) or S-net (Uehira et al. , 2016) might be a useful approach. For the 1998 PNG event, the tsunami source was about 20 km from the shoreline. This distance is shorter than the 30 km spacing between S-net sensors, so an array of similar dimensions to the S-net system would be too sparse for direct detection of tsunami waves such as those of the 1998 PNG tsunami with plural sensors. Other technologies with potential for direct detection of tsunami waves are tsunami radar (Barrick , 1979) and a fine barometer (Arai et al. , 2011), with which a tsunami is detected by reflection of electromagnetic waves or atmospheric pressure changes.

**5 Conclusions**

We investigated whether the tsunami-generating submarine landslide that occurred about 13 min after the 1998 PNG earthquake could be identified in the coda waves of the seismic data for the periods close to the landslide duration. We constructed synthetic seismograms to represent the seismic signature of the landslide and compared them to the seismic data recorded after the earthquake, with particular attention to the period around 13 min after the earthquake. We found small seismic phases possibly from the landslide. However, those phases were difficult to be recognized as an indication of a disastrous tsunami. Other methods are needed to provide data for early warnings of tsunamis generated by submarine landslides of similar size (or smaller) to the one that generated the 1998 PNG tsunami. Networks of ocean-bottom tsunami gauges, similar to those provided by the DONET and S-net arrays in Japan, are among the likely candidates for this approach.

*Data availability.* Seismic data from the Jayapura seismic station are available at http://ohpdmc.eri.u-tokyo.ac.jp/. Seismic data from other seismic stations shown in this study are available at https://www.iris.edu/hq/.

*Author contributions.* AKa analyzed the observed and synthetic seismic records and compiled most of the paper. YY suggested input to the methodology for construction of the synthetic records used in this study. KN researched previous studies of the 1998 PNG earthquake and tsunami. KF undertook preliminary research on the observed seismic records. MT, KT, TN, and AKo contributed to the research into previous studies of tsunamis caused by landslides and participated in related discussions.

5   *Competing interests.* The authors declare that they have no conflicts of interest directly relevant to the content of this article.

*Acknowledgements.* We used seismic data archived at the Earthquake Research Institute (The University of Tokyo) and at the Incorporated Research Institutions for Seismology. We used a computer program developed by Prof. Minoru Takeo to calculate synthetic records. We are grateful to Prof. Fryer, an anonymous reviewer, Prof. Tappin, and Prof. Okal for their thoughtful comments.

[revised manuscript text omitted]

---

## Author Comment (AC4) · 24 Jan 2019

Dear Editor and the reviewers of NHESS,

We summarize the responses to the comments from reviewers and other researchers, and show tentative revised manuscript as the supplement. The comments are categorized in several kinds below.

**1   Signals of high-frequency**

There were comments that the 1998 PNG landslide had been detected with hydrophones and high-frequency seismic data.

**Reviewer #1 (Prof. Fryer)** Fourth, Katsumata, et al. ignore hydrophones in their discussion of potential detection systems and instead suggest direct detection of the tsunami via pressure gauges like S-net. But direct detection is intrinsically slow because you have to wait for the tsunami to reach your sensor. Since sound waves in the ocean travel faster than the tsunami, hydrophones potentially provide more warning time and would therefore be superior. Again, the Caplan-Auerbach paper is relevant here.

**Prof. Tappin** Fryer identifies missing references and the potential for T-phase warning which could provide an alternative approach.

**Reviewer #2** First, Katsumata et al are trying to find the signature of the submarine slide in the seismic record, without mentioning and describing in the figures that 4 aftershocks were identified during the 22 Minutes following the main shock (Synolakis 2002). In Figure 3 JAY record shows that waves of the two largest aftershocks are arriving at 09:09.30 and 09:10:30 (not mentioned by the authors). Synolakis specified that one of the aftershocks at 09:02 mb 4.4 could correspond to the submarine landslide. In Figure 1 an additional record filtered 0.1- 1 s would probably help to show the high frequency waves of this aftershock, in fact the slide. Long duration of main shock ($> 2$-5 min) and aftershocks occurring in the tens of minutes after the main shock could definitively masked the waves generated by the submarine landslide generated in the 10-20 minutes following the quake. But as this event was identified and located by seismic waves picking and measurement, signal should be visible on the JAY record at higher frequency ($> 1$ Hz).

**Reviewer #2** Conclusion : Katsumata et al finally demonstrate that the synthetic record obtained for JAY seismic station doesn't match with the observed record in the specific band (50-100s). JAY record shows that no signal is visible in the bandwidth of 0.5s to 100s, 13 minutes after the quake, when the slide waves are expected. The conclusion of the authors is not relevant : other type and parameters of the slide could be modeled to compute synthetic records and compare with JAY record in higher frequency band (0.1 - 1s). Detect, identify and warn a tsunami due to submarine or aerial slide following large earthquake is definitively a complex challenge, essentially because of the duration of the quake and also the number and magnitude of aftershocks. As mentioned by Katsumata et al., S-net and DONET equipped with accelerometers, seismometers and pressure sensors are the most likely candidates to detect and warn submarine landslide. Nevertheless seismic arrays and seismic stations located closer to the slide ($< 100$ km) could be able to detect slide waves. In addition, hydroacoustic arrays (Synolakis) and coastal seismic station located on islands close to the epicenter could also help to detect T phase generated by the quakes and those generated by the slide. This paper needs major revision.

**Prof. Tappin** The anonymous reviewer (RC2) undoubtedly identifies a fundamental flaw in the analysis. The PNG 09.02 'seismic' event is unique and was identified by Synolakis et al. as reflecting the slump movement, not a seismic event. The modelled and observed signals do not match. The authors have to go back and revisit the frequency signal at this time, and address this before the paper can be published – it is fundamental to the papers conclusions and the possibilities for tsunami warning from submarine landslides.

**Prof. Okal** The main argument of the paper is that there is no detectable seismic signature to the landslide which generated the catastrophic PNG tsunami of 17 July 1998. This statement directly contradicts the work of Okal [2003], in which I presented (on Figure 3) and discussed in detail the record of the landslide at the

same station JAY allegedly studied by the authors. It is clear that they used the wrong (very low-frequency) filters, and thus missed the signal. They do not justify working in such inadequate frequency bands, and completely ignore the detailed analysis of seismic and hydroacoustic phases which went into my 2003 paper.

(Response)

- We added descriptions about detections of the landslide and aftershocks by hydrophones and high-frequency seismic wave. However, We considered that size estimation is indispensable for tsunami warning purpose. Size of landslide is an indispensable factor to estimate tsunami height. Passband of instruments should cover the process duration to estimate the size of the event properly. Frequency ranges of hydrophones and high-frequency seismic records do not cover the frequency related to the duration of the landslide. We missed to describe importance of landslide size estimation for tsunami warning purpose in the previous manuscript. We added description in "Introduction" to claim that hydrophone is not useful for estimation of landslide size. The duration of the waves would reflect the duration of the landslide. However it is not directly connected to the mass.

**2  Analysis method**

**Reviewer #2**  Second, the synthetic records obtained by modeling by Katsumata et al, for the closest station JAY, are of much bigger than the waves on records, in the 50-100s band. As mentioned by Fryer, the slide could be a slump type, or debris avalanche type, and in addition, the rheology parameters could vary extremely. The conclusion is, because no signal is visible in the bandwidth 50-100 s on the JAY station record, the hypothesis of the synthetic source and propagation performed by the authors is probably not correct. Katsumata et al should performed

other synthetic records, knowing that in the Figure 2 shows that, at JAY station, in the bandwidth 0.5s to 100s, no clear signal is visible at the theoretical arrival time of the waves of the slide.

- Analatical solution for a homogeneous unbounded media shows that far field displacement is proportional to the force amplitude (e.g., Aki and Richards, 2002). Whereas complex process may affect the seismic records and total energy may be changed extremely according to travel distance, the peak force acting on the ground should be constrained by the total mass and its acceleration.

  The calculation procedure was checked with the result of Takeo (1990, JGR). It is true that the synthetic amplitude in Fig. 5 is too large compared with the observed records. Seismic phases are not recognized either at GUMO, CTAO, WRAB, and DAV. A simple explanation for those would be that the assumed force might be too large. We do not insist on the correctness of the assumption. Rather our conclusion is that detection of landslide with long-period seismic wave is difficult after a big earthquakes.

**Reviewer #2** Third, other processing methods exist to help to identify waves visually or by signal processing : computation of spectrograms is one of the efficient method, and computation of polarization parameters of waves.

- We added spectrograms in Figure 2.

**3   Landslide type**

**Reviewer #1 (Prof. Fryer)** First, the paper treats all submarine landslides as if they are the same, but landslides have a broad range of characteristics which should at
least be mentioned. The landslide types most important in generating tsunamis are slumps and debris avalanches. A slump is a landslide in which a coherent block of material slides downslope on a rotational slip surface. The event is of relatively short duration (a few tens of seconds) and downslope motion is relatively small, so remote detection of the slope failure is going to be a challenge. The PNG tsunami came from such a source. A debris avalanche, by contrast, involves complete disintegration of the sliding body, motion lasts for a long time (conceivably several minutes), and both downslope motion and runout at the base of the slope are large. The St. Helens landslide the authors refer to was a debris avalanche. Even if we cannot warn of slump-generated tsunamis, the larger signals from debris avalanches should allow us to warn of those events.

**Prof. Tappin** To confirm, the PNG landslide was a slump. I mapped it.

- We added description about various types of submarine mass failure referring to Schwab et al. (1993), and mentioned that the 1998 PNG event had a relatively short travel distance.

  It is considered that the amplitudes of seismic waves are proportional to the peak force. Even if the duration of mass motion is long, the amplitude of the seismic waves would not be different for the same size of forces. We added this description in the section of synthetic seismogram.

**Correspondence:** Akio Katsumata (akatsuma@mri-jma.go.jp)

**Abstract.** On 17 July 1998, a tsunami caused serious damage on the northern coast of Papua New Guinea about 20 min after the mainshock of an $M_w$ 7.0 earthquake. The tsunami has been attributed to a submarine landslide that occurred about 13 min after the mainshock because its arrival at the coast was too late and its height too great to be the direct result of the fault slip of the earthquake. Bathymetric data recorded after the tsunami revealed an amphitheater-like structure that was consistent with a recent submarine landslide. Most current tsunami warning systems are based on analysis of the early arrivals of seismic waves generated by an earthquake. In this study we investigated whether evidence of the landslide could be identified in the coda waves of long period recorded after the mainshock. Seismic waves of period ranges close to the landslide duration are considered to be helpful for its size estimation. Based on previous studies of the tsunami source, we constructed synthetic seismograms to represent the submarine landslide and compared them to the observed coda waves of the preceding earthquake, with particular attention to the period around 13 min after the mainshock. We found phases possibly corresponding to the landslide event. However, they were easily covered with coda waves from the mainshock. We concluded that the 1998 landslide was too small to be evident in the coda waves following the magnitude 7 earthquake.

*Copyright statement.* TEXT

**1 Introduction**

A tsunami struck the north coast of Papua New Guinea (PNG) on 17 July 1998. The height of the tsunami was more than 10 m and its passage resulted in the deaths of more than two thousand people (e.g., Kawata et al., 1999; Tappin et al., 1999; Synolakis et al., 2002). The tsunami struck the coast about 20 min after an earthquake (Kawata et al., 1999) of $M_w$ 7.0 (GCMT, Dziewonski et al., 1981; Ekström et al., 2012). The arrival time of the tsunami was too late and its height too great for it to have been a direct result of the fault slip of the earthquake. Moreover, bathymetric data recorded after the earthquake revealed an amphitheater-like structure consistent with recent submarine slumping about 20 km off the coast (Tappin et al., 1999). It was therefore presumed that the tsunami was caused by a submarine landslide (e.g., Tappin et al., 1999; Heinrich et al., 2000; Synolakis et al., 2002). Numerical simulations of the landslide (e.g., Tappin et al., 1999; Watts et al., 2003; Lynett et al.,

2003) provided landslide masses of 6.4 km$^3$ (Tappin et al., 2008), 4 km$^3$ (Heinrich et al., 2000), and 9 km$^3$ (Watts et al., 2003).

Various types of submarine mass failures have been recognized so far, and some mass flows traveled hundreds of kilometers (e.g., Schwab et al. , 1993). The mass motion distance of the 1998 PNG event was estimated at a relatively short one (e.g., 766 m by Watts et al., 2005; 980 m by Tappin et al., 2008), and was a slump type.

Early warnings of impending tsunamis are usually issued based on analyses of the early arrivals of seismic waves generated by earthquakes. If the submarine landslide after the 1998 PNG earthquake had been detected in the seismic data recorded immediately after the earthquake, it might have been possible to provide advanced warnings to PNG residents of the imminent tsunami. In this study, we investigated whether the seismic signature of the submarine landslide was present in the coda waves of the seismic data recorded after the mainshock of the 1998 earthquake. The 1998 PNG landslide was identified in a hydrophone record (Synolakis et al., 2002; Okal, 2003) and a high-frequency seismic record (Okal, 2003). Caplan-Auerbach et al. (2001) reported hydrophone records from submarine landslides at Kilauea volcano. It is necessary to estimate size of a landslide as well as its source type for tsunami warning purpose. To estimate source size from observed amplitudes, frequency range of the records should cover the process duration for a proper estimation (e.g., Aki, 1967). The duration of the 1998 PNG landslide was estimated at several tens to 100 s (Watts et al., 2003; Tappin et al., 2008). The passband of the hydrophones (e.g., 1–40 Hz of the PMEL hydrophone array, Caplan-Auerbach et al., 2001; 1–30 Hz at Wake Island, Okal, 2003) seems to be insufficient to estimate the size of landslides. Whereas the duration of the hydrophone waves would have reflected the duration of the landslide (Synolakis et al., 2002; Okal, 2003), the mass size and the duration are not directly related to each other. We focused on seismic records of period ranges around several tens of seconds.

**2  Observed seismic records**

The broadband seismic data recorded during the $M_w$ 7.0 earthquake at 08:49:13 (UTC) that preceded the 1998 tsunami has been archived at both the Earthquake Research Institute (ERI; The University of Tokyo) and the Incorporated Research Institutions for Seismology (IRIS; headquarters in Washington DC). The two seismic stations from which data were mainly used in this study were 146 km (Jayapura station, JAY) and 918 km (Port Moresby station, PMG) from the epicenter (Fig. 1).

Synolakis et al. (2002) calculated that the submarine landslide occurred 13 min after the mainshock on the basis of hydrophone data recorded at Wake Island. Heidarzadeh and Stake (2015) used tide gauge data to estimate that the landslide occurred 12–17 min after the mainshock. Synolakis et al. (2002) pointed out that seismic signal corresponding to the landslide could not be recognized at PMG. Okal (2003) showed a highpass-filtered seismic record (> 1 Hz) obtained at JAY station, and attributed an event at 09:02 (UTC) to the landslide. The characteristic time of the 1998 PNG landslide was estimated at several tens of seconds (32 s by Watts et al., 2003; 44 s by Tappin et al., 2008). Seismic signal is expected to be seen in this period range. Records at JAY show no significant events at around 13 min after the mainshock in the four frequency bands considered and spectrographs of 0.01-1 Hz (Fig. 2). Aftershocks occurred at 09:02:06 ($m_b$ 4.4, UTC) and 09:06:03 (no magnitude reported, UTC). The event at 09:02:06 was considered to be directly related to the submarine landslide (Synolakis et al., 2002;

Okal, 2003). Those earthquakes are not evident in the frequency and amplitude range of Fig. 2. Whereas any significant events are not seen in the records at six stations around source area (Fig. 3), small phases are recognized about 13 min after the S-arrival in the 50–100 s band of horizontal components at JAY and PMG (bold parts of seismic waves in Fig. 3). The particle motions of the bold parts were oriented to north-south at JAY and northeast-southwest at PMG, which were roughly consistent

5 with those of the Love waves from the source area. However, there were not such events in the records at the other stations (Fig. 3).

**3 Comparison of synthetic seismograms representing the 1998 submarine landslide with coda waves from the 1998 earthquake**

We estimated the amplitudes of the seismic waves generated by the 1998 landslide on the basis of the work of Watts et al.

10 (2003), as described below. Relatively large amplitudes were expected based on the model by Watts et al. (2003) because of its relatively large assumed mass.

When a landslide starts, the change in the force that has supported the landslide mass generates seismic waves. The force ($F$) generated by a landslide is defined by $F = ma$, where $m$ is the mass of the landslide and $a$ is its acceleration. Watts et al. (2003) assumed the landslide mass to be a half ellipsoid and, on the basis of the parameters given in Table 1, estimated its

15 volume to be 9 km$^3$ ($\frac{\pi wbT}{6}$) and its mass to be $19 \times 10^{12}$ kg. Using this mass and the initial acceleration $a_0$ (0.36 m s$^{-2}$; Table 1) of Watts et al. (2003) gives an estimated force of $7 \times 10^{12}$ N. The volume determined by Watts et al. (2003) (9 km$^3$) is larger than those of Heinrich et al. (2000) (4 km$^3$), Synolakis et al. (2002) (4 km$^3$), and Tappin et al. (2008) (6.4 km$^3$), and the acceleration of 0.36 m s$^{-2}$ is smaller than that used by Tappin et al. (2008) (0.47 m s$^{-2}$). The force and volume estimated here for the 1998 PNG landslide are comparable to those estimated by Kanamori and Given (1982) for the landslide associated

20 with the 1980 Mount St. Helens eruption ($10^{13}$ N and 2.5 km$^3$, respectively).

Amplitudes of far-field displacement of seismic waves in a homogeneous unbounded medium are theoretically proportional to the amplitude of the acting force (e.g., Aki and Richards, 2002). For a case of a longer-duration mass failure, the total energy would become greater. Even for such a case, the amplitudes of long-period seismic waves should be constrained by the peak of the force.

25 The time history of a landslide must be assumed in order to estimate the amplitude of the resultant seismic wave. Watts et al. (2003) assumed a characteristic time of 32 s. Tappin et al. (2008) estimated that the sliding process lasted about 100 s. We assumed that the sliding process was completed within several tens of seconds with an initial acceleration stage followed by a deceleration stage (Ekström et al., 2012) as shown in Fig. 4), and that the total impulse would not have been balanced because part of the deceleration stage would be affected by interaction of the sliding mass with sea water. Since it was estimated that

30 the landslide occurred inside the existing amphitheatre (Synolakis et al., 2002; Okal, 2003; Sweet and Silver, 2003), it may have stopped with a large acceleration. Here we assume the same peak values of force for acceleration and deceleration. The curves for each of the four intervals defined on Fig. 4 were expressed as trigonometric functions (Table 2). Five different time histories were considered (Table 3) to see difference due to duration variation. The peaks of $\int F \mathrm{d}t/m$ range from 3 to 11 m s$^{-1}$.

The value 11 m s$^{-1}$ of case (c) (Table 3) is close to $u_{max}$ of Table 1. We constructed synthetic seismic records (Figs. 5 and 6) by applying the method of Takeo (1985) and using the seismic velocity model given in Table 4. In Figs. 5 and 6, azimuthal relationship between the force and the component are assumed so that largest radiation is expected for the direction.

The amplitudes of of our synthetic records are comparable to or larger than those of the seismic waves recorded after the mainshock of the $M_w$ 7.0 earthquake in the 50–100 s passband (Figs. 5 and 6). The phases indicated with bold lines could be the seismic waves from the landslide which caused the disastrous tsunamis. Whereas the ratios of the observed amplitudes to the synthetics at JAY were generally small, those at PMG were not so small. It is difficult to retrieve a consistent result about the source parameters from this comparison. The amplitudes amplitude of the phase was smaller than the successive seismic wave at JAY.

Ekström et al. (2012) presented scaling relationship of landslides. The force of $7 \times 10^{12}$ N corresponds to 5.8 of surface wave magnitude according to the scaling relation. A vertical displacement of 0.05 mm is expected at a epicentral distance of 2000 km and at a period of 20 s for this surface wave magnitude (Vaněk et al., 1962). The amplitude of 0.05 mm is comparable to the observed amplitudes of coda waves at stations of epicentral distance around 2000 km for a passband around 20 s (Fig. 7). It is unlikely to recognize the occurrence of the landslide based on the long-period seismic records after the large earthquake.

**4   Discussion**

Several studies have shown that seismic waves generated by landslides can be detected. The landslide associated with the 1980 eruption of Mount St. Helens (Kanamori and Given, 1982; vol. 2.5 km$^3$) is the largest of these. Yamada et al. (2012) analyzed seismic waves generated by landslides (vols. of up to $13.6 \times 10^6$ m$^3$) caused by heavy rain in Japan. Li et al. (2017) used seismic data to analyze the dynamic process of a landslide (vol. $5 \times 10^6$ m$^3$) in southwest China. Landslides in Greenland that caused tsunamis in 2000 (Dahl-Jensen et al., 2004; underwater vol. $30 \times 10^6$ m$^3$) and 2017 (Chao et al., 2018; vol. 35–51 $\times 10^6$ m$^3$) were recorded by seismometers. For the 2000 Greenland event, Dahl-Jensen et al. (2004) estimated the surface-wave magnitude of the event to be 2.3. Ekström and Stark (2013) showed scaling relationship of large landslides.

In cases such as the 1998 PNG tsunami, where the submarine landslide that caused it occurred some minutes after an earthquake, the long-period seismic signature of the landslide can be masked by the seismic waves generated by the earthquake. In 1908, after a magnitude 7.1 earthquake, the southern Italian coast was struck by a 5–10 m tsunami (?, ?Salamon et al., 2007; Papadopoulos et al., 2014) that Billi et al. (2008) attributed to a submarine landslide. The event in Italy in 1908 appears to be similar to the 1998 PNG earthquake and tsunami.

Because some onshore earthquakes of magnitude about 7 or greater are known to have caused landslides, it is likely that offshore earthquakes of similar magnitude cause submarine landslides. Kodaira et al. (2012) suggested that a submarine land-slide might have been caused by the 2011 Tohoku-oki earthquake ($M_w$ 9.0). Tappin et al. (2014) modeled a tsunami source with a landslide to explain tsunami heights along the coast of Sanriku district of the 2011 Tohoku-oki earthquake. However, the seismic and tsunami signatures of such landslides are likely masked by the responses to the fault motions of the earthquakes.

An extraordinarily large landslide mass would be required to prevent its signatures from being overwhelmed by the responses of the fault motion of a magnitude 7 or greater earthquake.

It is considered that long periods of seismic inactivity or quick sedimentation (Sawyer et al., 2017) is required for the accumulation on submarine slopes of sufficient sediment to generate a landslide large enough to cause a tsunami of comparable size to the 1998 PNG tsunami. The Mediterranean is  a region where seismic activity is low enough that many of the known tsunamis may have been caused by  landslides (Salamon et al., 2007; Papadopoulos et al., 2014). Because of the greater seismicity, such conditions do not exist in southeast Asia. It is plausible  there that heavy rainfall and rapid deposition of terrigenous sediment offshore might contribute to the occurrence of submarine landslides, including the 1998 PNG landslide, despite their location in a seismically active region. A few tsunamis in Japan, where tectonic activity is very high, have also been attributed to submarine landslides (e.g., Baba et al., 2012; Baba et al., 2017).

Chao et al. (2018) considered the use of seismic records to detect landslides in order to provide early warnings of impending tsunamis. However, as we have demonstrated here, the long-period seismic signature of large submarine landslides can be overwhelmed by the seismic coda waves generated by earthquakes, so it can be difficult to detect submarine landslides soon after large earthquakes by this method. An alternative method of providing early warnings for tsunamis generated by submarine landslides is needed. Monitoring by networks of ocean-bottom tsunami gauges such as those of DONET (Kawaguchi et al., 2015) or S-net (Uehira et al., 2016) might be a useful approach. For the 1998 PNG event, the tsunami source was about 20 km from the shoreline. This distance is shorter than the 30 km spacing between S-net sensors, so an array of similar dimensions to the S-net system would be too sparse for direct detection of tsunami waves such as those of the 1998 PNG tsunami with plural sensors. Other technologies with potential for direct detection of tsunami waves are tsunami radar (Barrick, 1979) and a fine barometer (Arai et al. , 2011), with which a tsunami is detected by reflection of electromagnetic waves or long period atmospheric pressure changes.

**5  Conclusions**

We investigated whether the tsunami-generating submarine landslide that occurred about 13 min after the 1998 PNG earthquake could be identified in the coda waves of the seismic data for the periods close to the landslide duration. We constructed synthetic seismograms to represent the seismic signature of the landslide and compared them to the seismic data recorded after the earthquake, with particular attention to the period around 13 min after the earthquake. We found small seismic phases possibly from the landslide. However, those phases were difficult to be recognized as an indication of a disastrous tsunami. Other methods are needed to provide data for early warnings of tsunamis generated by submarine landslides of similar size (or smaller) to the one that generated the 1998 PNG tsunami. Networks of ocean-bottom tsunami gauges, similar to those provided by the DONET and S-net arrays in Japan, are among the likely candidates for this approach.

*Data availability.* Seismic data from the Jayapura seismic station are available at http://ohpdmc.eri.u-tokyo.ac.jp/. Seismic data from other seismic stations shown in this study are available at https://www.iris.edu/hq/.

*Author contributions.* AKa analyzed the observed and synthetic seismic records and compiled most of the paper. YY suggested input to the methodology for construction of the synthetic records used in this study. KN researched previous studies of the 1998 PNG earthquake and tsunami. KF undertook preliminary research on the observed seismic records. MT, KT, TN, and AKo contributed to the research into previous studies of tsunamis caused by landslides and participated in related discussions.

*Competing interests.* The authors declare that they have no conflicts of interest directly relevant to the content of this article.

*Acknowledgements.* We used seismic data archived at the Earthquake Research Institute (The University of Tokyo) and at the Incorporated Research Institutions for Seismology. We used a computer program developed by Prof. Minoru Takeo to calculate synthetic records. We are
10 grateful to Prof. Fryer, an anonymous reviewer, Prof. Tappin, and Prof. Okal for their thoughtful comments.

[revised manuscript text omitted]